# Einstein Aggregation Operators under Bipolar Neutrosophic Environment with Applications in Multi-Criteria Decision-Making

**Muhammad Jamil** [1]**, Farkhanda Afzal** [1]**, Ali Akgül** [2,3,*]**, Saleem Abdullah** [4]**, Ayesha Maqbool** [1]**, Abdul Razzaque** [1]**, Muhammad Bilal Riaz** [5] **and Jan Awrejcewicz** [6,*]

[1] Military College of Signal, National University of Sciences and Technology, Islamabad44000,
[2] Department of Mathematics, Art and Science Faculty, Siirt University, 56100 Siirt, Turkey
[3] Department of Mathematics, Mathematics Research Center, Near East University, Near East Boulevard, Mersin 10, 99138 Nicosia, Turkey
[4] Department of Mathematics, Abdul Wali Khan University, Mardan 23200,
[5] Faculty of Technical Physics, Information Technology and Applied Mathematics, Lodz University of Technology 90-924 Lodz, Poland
[6] Faculty of Mechanical Engineering, Lodz University of Technology, 90-924 Lodz, Poland
[*] Correspondence: aliakgul@siirt.edu.tr (A.A.);jan.awrejcewicz@plodz.pl (J.A.)

**Abstract:** In this article, we introduce bipolar neutrosophic (BN) aggregation operators (AOs) as a revolutionary notion in aggregation operators (AOs) by applying Einstein operations to bipolar neutrosophic aggregation operators (AOs), with its application related to a real-life problem. The neutrosophic set is able to drawout the incomplete, inconsistent and indeterminate information pretty efficiently. Initially, we present essential definitions along with operations correlated to the neutrosophic set (NS) and its generalization, the bipolar neutrosophic set (BNS). The Einstein aggregation operators are our primary targets, such asthe BN Einstein weighted average (BNEWA), BN Einstein ordered weighted average (BNEOWA), BN Einstein hybrid average (BNEHA), BN Einstein weighted geometric (BNEWG), BN Einstein ordered weighted geometric (BNEOWG) and BN Einstein hybrid geometric (BNEHG), as well as their required properties. The most important benefit of using the suggested approaches is that they provide decision-makers with complete sight of the issue. These techniques, when compared to other methods, provide complete, progressive and precise findings. Lastly, by means of diverse types of newly introduced aggregation operators and a numerical illustration by an example, we suggest an innovative method to be used for multi-criteria community decision-making (DM). This illustrates the utility and applicability of this new strategy when facing real-world problems.

**Keywords:** aggregation operator; decision-making; BNEWA; BNEOWA; BNEHA; BNEWG; BNEOWG; BNEHG

## 1. Introduction

In the modern age of managerial decision-making (DM), knowledge frequently remains incomplete, undetermined and incompatible. L. A. Zadeh was the first to propose the fuzzy set theory [1], which deals with uncertainty and has applications in a wide range of modern fields of present and future society. Yet, fuzzy set has a problem in that it can simply convey the value of membership butnot non-membership. To address this, Atanassov [2] developed the intuitionistic fuzzy set (IFS), as well as associated theory, in order to summarizethe concept of fuzzy set.A pair of membership values is used to represent each IFS element, truth-membership $\Im(\chi)$ and a non-membership value (falsity-

membership) $f(\chi)$, as well as satisfy the condition $\Im(\chi), f(\chi) \in [0,1]$ with $0 \leq \Im(\chi) + f(\chi) \leq 1$. IFS can only handle incomplete data, meaningindeterminate data are not an option.

Florentin Smarandache [3] developed the novel concept of NS, which adds an indeterminacy membership value $I(\chi)$ to IFS. NS has a strong ability to deal with knowledge that is imperfect, uncertain and contradicting.

When $\Im(\chi) + I(\chi) + f(\chi) < 1$, the information acquired is indeterminate.

When $\Im(\chi) + I(\chi) + f(\chi) > 1$, this represents inconsistency in an NS.

Wang and others devised the single-valued neutrosophic set (SVNS) to deal with real-world situations [4], with the conditions $\Im(\chi), I(\chi), f(\chi) \in [0,1]$ and $0 \leq \Im(\chi) + I(\chi) + f(\chi) \leq 3$. Ye [5] proposed a method for comparing SVNS, and defined the correlation coefficient.Wang et al. formulated an interval-valued neutrosophic set [6] and broadened the values of truth, indeterminacy and false memberships, which are all between 0 and 1.

For researchers, aggregation operators (AOs) are extremely important.Since its introduction, several scientists [7–14] have contributed significantly to the theory development of IFS.Xu et al. [15] introduced the concept of distinct IF aggregation operators based on IFS (AOs).Wang et al. and Zhao et al. [16,17] created Einstein aggregation operators (AOs). The Einstein t-norm typically gives the same smooth approximations as the product and sum of algebra.

The algebraic t-norm (product $\otimes$ ) and t-conorm (sum $\oplus$ ), respectively, are as below:

$$T(a,b) = a \otimes b = ab$$
$$T^*(a,b) = a \oplus b = a + b - ab$$

The Einstein operations t-norm (product $\otimes$ ) and t-conorm (sum $\oplus$ ),respectively, are as below:

$$T(a,b) = a \otimes b = \frac{ab}{1 + (1-a)(1-b)}$$
$$T^*(a,b) = a \oplus b = \frac{a+b}{1+ab}$$

The bipolar fuzzy set (BFS) [18–20] has emerged as a new technique to deal with vagueness in DM situations.The bipolar fuzzy set's membership degree varies between −1 and 1. Positive as well as negative membership degrees are available in BFS. BFSs are incredibly beneficial for several fields of study, as well as DM [21,22]. Gul [23] defined bipolar averaging and geometric aggregations operators (AOs). The bipolar neutrosophic set was created by Irfan and others [24,25] by using fundamental operations and a comparison mechanism.[26,27].Jamil and others created aggregation operators (AOs) based on BN values and applied them to DM problems. Heronian mean aggregation operators were developed by Fan and others [28]. Abdullah et al. introduced the idea of a bipolar soft set and its application to decision-making [29]. Jafar and other developed a bipolar neutrosophic soft set and applied itto decision-making [30]. Ali et al. introduced complex fuzzy set and its properties [31]. Broumi et al. introduced bipolar complex fuzzy set and its aggregation operators [32]. Jamil and others applieda multi-criteria decision-making approach to the bipolar neutrosophic set [33].

Regardless of the available information, there isa lot of literature related to the subject. The following features of the bipolar neutrosophic set inspired the present research

team to perform a systematic and in-depth inquiry into decision analysis. Our findings are given below:

SVNSs deal with ambiguous details easily. This set combines the generalization of prior sets such as the classical set, the FS set and the IFS set.BFSs are highly effective for dealing withunpredictable,unexpected real-world circumstances because these can handle both positive as well as negative membership values.

The current study's main and most important goals were as follows:

- To suggest different bipolar neutrosophic Einstein AOs as well as desired properties to study;
- Based on BNN,establish a multi-criteria DM approach in the direction of real life problem-solving;
- Give a numerical description of amulti-criteria DM example.

The remainder of the paper is organized at follows. The second segment presents a basic definition as well as its associated properties.The BNEWA and BNEWG aggregation operators are introduced in thethird section.These innovative AOs are applied to multi-criteria decision-making in section four, and a numerical example is also presented.Finally, we offer comparative research, as well as some concluding thoughts, in section five.

## 2. Preliminaries

In this section, we provide fundamental definitions related to neutrosophic set theory. We define different fuzzy sets, BNS, scorefunctions, accuracyfunctions, and certainty functions, as well as the Einstein operations.

**Definition 1 [3].** *Consider R to represent a fixed set. Then, the neutrosophic set N is defined as:*

$$N = \left\{ \left( \chi, \mathfrak{I}(\chi), \mathrm{I}(\chi), f(\chi) \right) \mid \chi \in R \right\}$$

*Now, the mapping for membership functions of truth, indeterminacy and falsity are* $\mathfrak{I}: N \to Q$ , $\mathrm{I}: N \to Q$ *and* $f: N \to Q$ , *respectively; here,* $Q = \left] 0^-, 1^+ \right[$ *and* $0^- \le \mathfrak{I}(\chi) + \mathrm{I}(\chi) + f(\chi) \le 3^+$ .

**Definition 2 [4].** *Consider P to represent a fixed set; the single-valued neutrosophic set (SVNS) of A is stated as:*

$$A_{NS} = \left\{ \left( \chi, \mathfrak{I}(\chi), \mathrm{I}(\chi), f(\chi) \right) \mid \chi \in P \right\}$$

*Now, here, the mapping for membership functions of truth, indeterminacy and falsity are* $\mathfrak{I}: A_{NS} \to L$ , $\mathrm{I}: A_{NS} \to L$ *and* $f: A_{NS} \to L$ , *respectively; here,* $L = \left[ 0, 1 \right]$ *and there is the condition* $0 \le \mathfrak{I}(\chi) + \mathrm{I}(\chi) + f(\chi) \le 3.$

*Some fundamental operations related to single-valued neutrosophic sets (SVNS) are listed below:*

$$A_{NS} = \left\{ \left( \chi, \mathfrak{I}_{A_{NS}}(\chi), \mathrm{I}_{A_{NS}}(\chi), f_{A_{NS}}(\chi) \right) \mid \chi \in P \right\}, \quad \mathrm{B}_{NS} = \left\{ \left( \chi, \mathfrak{I}_{B_{NS}}(\chi), \mathrm{I}_{B_{NS}}(\chi), f_{B_{NS}}(\chi) \right) \mid \chi \in P \right\},$$

*The subset* $A_{NS} \subseteq B_{NS}$ *is represented as*

$$\mathfrak{I}_{A_{NS}}(\chi) \le \mathfrak{I}_{B_{NS}}(\chi), \mathrm{I}_{A_{NS}}(\chi) \ge \mathrm{I}_{B_{NS}}(\chi), f_{A_{NS}}(\chi) \ge f_{B_{NS}}(\chi).$$

*$A_{NS} = B_{NS}$ is represented as*

$$\mathfrak{I}_{A_{NS}}(\chi) = \mathfrak{I}_{B_{NS}}(\chi), \mathrm{I}_{A_{NS}}(\chi) = \mathrm{I}_{B_{NS}}(\chi), f_{A_{NS}}(\chi) = f_{B_{NS}}(\chi).$$

The complement $A_{NS}^{'}$ is

$$A_{NS}^{'} = \left\{ \left( \chi, f_{A_{NS}}(\chi), 1 - \mathrm{I}_{A_{NS}}(\chi), \mathfrak{I}_{A_{NS}}(\chi) \right) \mid \chi \in P \right\}$$

$A_{NS} \cap B_{NS}$ is represented as

$$A_{NS} \cap B_{NS} = \left\{ \left( \chi, \min\left\{ \mathfrak{I}_{A_{NS}}(\chi), \mathfrak{I}_{B_{NS}}(\chi) \right\}, \max\left\{ \mathrm{I}_{A_{NS}}(\chi), \mathrm{I}_{B_{NS}}(\chi) \right\}, \max\left\{ f_{A_{NS}}(\chi), f_{B_{NS}}(\chi) \right\} \right) \mid \chi \in P \right\}$$

The union is defined by

$$A_{NS} \cup B_{NS} = \left\{ \left( \chi, \max\left\{ \mathfrak{I}_{A_{NS}}(\chi), \mathfrak{I}_{B_{NS}}(\chi) \right\}, \min\left\{ \mathrm{I}_{A_{NS}}(\chi), \mathrm{I}_{B_{NS}}(\chi) \right\}, \min\left\{ f_{A_{NS}}(\chi), f_{B_{NS}}(\chi) \right\} \right) \mid \chi \in P \right\}$$

**Definition 3 [5].** *Consider two single-valued neutrosophic numbers (SVNNs)* $u_1 = \left( \mathfrak{I}_1, \mathrm{I}_1, f_1 \right)$ *and* $u_2 = \left( \mathfrak{I}_2, \mathrm{I}_2, f_2 \right)$. *The following are the distinct basic operations for SVNNs:*

$$u_1 + u_2 = \left( \mathfrak{I}_1 + \mathfrak{I}_2 - \mathfrak{I}_1 \mathfrak{I}_2, \mathrm{I}_1 \mathrm{I}_2, f_1 f_2 \right);$$
$$u_1 \cdot u_2 = \left( \mathfrak{I}_1 + \mathfrak{I}_2, \mathrm{I}_1 + \mathrm{I}_2 - \mathrm{I}_1 \mathrm{I}_2, f_1 + f_2 - f_1 f_2 \right);$$
$$\lambda \left( u_1 \right) = \left( 1 - \left( 1 - \mathfrak{I}_1 \right)^{\lambda}, \left( \mathrm{I}_1 \right)^{\lambda}, \left( f_1 \right)^{\lambda} \right);$$
$$\left( u_1 \right)^{\lambda} = \left( \left( \mathfrak{I}_1 \right)^{\lambda}, 1 - \left( 1 - \mathrm{I}_1 \right)^{\lambda}, 1 - \left( 1 - f_1 \right)^{\lambda} \right);$$

*where* $\lambda > 0$.

**Definition 4 [5].** *Let* $u_1 = \left( \mathfrak{I}_1, \mathrm{I}_1, f_1 \right)$ *be a single-valued neutrosophic number (SVNN). Then, the score function* $s\left( u_1 \right)$ *is defined below:*

$$s\left( u_1 \right) = \frac{\left( \mathfrak{I}_1 + 1 - \mathrm{I}_1 + 1 - f_1 \right)}{3};$$

**Definition 5 [5].** *The accuracy function for an SVNN* $u_1 = \left( \mathfrak{I}_1, \mathrm{I}_1, f_1 \right)$ *is denoted by* $\left( u_1 \right)$, *defined below:*

$$\left( u_1 \right) = \left( \mathfrak{I}_1 - f_1 \right);$$

**Definition 6 [5].** *The certainty function for an SVNN* $u_1 = \left( \mathfrak{I}_1, \mathrm{I}_1, f_1 \right)$ *is denoted by* $c\left( u_1 \right)$, *defined below:*

$$c\left( u_1 \right) = \mathfrak{I}_1$$

**Definition 7 [5].** *Consider two SVNNs* $u_1 = (\mathfrak{I}_1, I_1, f_1)$ *and* $u_2 = (\mathfrak{I}_2, I_2, f_2)$. *The following is the relationship between them:*

i. *In condition* $s(u_1) > s(u_2)$, *then* $u_1$ *is greater than* $u_2$, *denoted by* $u_1 > u_2$;

ii. *In condition* $s(u_1) = s(u_2)$ *and* $a(u_1) > a(u_2)$, *then* $u_1$ *is greater than* $u_2$, *denoted by* $u_1 > u_2$;

iii. *In condition* $s(u_1) = s(u_2)$, $a(u_1) = a(u_2)$ *and* $c(u_1) > c(u_2)$, *in that case,* $u_1$ *is superior to* $u_2$, *denoted by* $u_1 > u_2$;

iv. *In condition* $s(u_1) = s(u_2)$, $a(u_1) = a(u_2)$ *and* $c(u_1) = c(u_2)$, *in that case,* $u_1$ *is equal to* $u_2$, *denoted by* $u_1 = u_2$;

**Definition 8 [21].** *Consider P as a universal set. Then, the bipolar fuzzy set (BFS) is stated below:*

$$F = \left\{ \left\langle \chi, \mathfrak{I}_F^+(\chi), f_F^-(\chi) \right\rangle \mid \chi \in P \right\}$$

*where* $\mathfrak{I}_F^+(\chi): F \to L^+$ *represents a positive membership function and the negative membership function is* $f_F^-(\chi): F \to K^-$, *where* $L^+ = [0, 1]$ *and* $K^- = [-1, 0]$.

**Definition 9 [25].** *Suppose A is a bipolar neutrosophic set contained by universal set P, stated as:*

$$A = \left\{ \left( \chi, \mathfrak{I}^+(\chi), I^+(\chi), f^+(\chi), \mathfrak{I}^-(\chi), I^-(\chi), f^-(\chi) \right) \mid \chi \in P \right\}$$

*Let* $\mathfrak{I}^+(\chi), I^+(\chi), f^+(\chi) = BN^+$ *as well as* $\mathfrak{I}^-(\chi), I^-(\chi), f^-(\chi) = BN^-$.

*Here,* $\mathfrak{I}^+(\chi), I^+(\chi), f^+(\chi)$ *are positive membership functions representing true, indeterminate and false for* $\chi \in P$, *and* $\mathfrak{I}^-(\chi), I^-(\chi), f^-(\chi)$ *represent true, indeterminate and false for negative membership functions. Now, when mapping* $BN^+: A \to L^+$ *and* $BN^-: A \to L^-$, *here,* $L^+ = [0, 1]$ *and* $L^- = [-1, 0]$. *This applies with the condition* $0 \leq \mathfrak{I}^+(\chi) + I^+(\chi) + f^+(\chi) + \mathfrak{I}^-(\chi) + I^-(\chi) + f^-(\chi) \leq 6$.

**Example 1.** *Let* $P = \{\chi_1, \chi_2, \chi_3\}$, *then*

$$A = \left\{ \begin{array}{l} (\chi_1, 0.1, 0.5, 0.3, -0.4, -0.5, -0.6), \\ (\chi_2, 0.2, 0.7, 0.4, -0.3, -0.6, -0.2), \\ (\chi_3, 0.4, 0.6, 0.7, -0.2, -0.3, -0.1) \end{array} \right\}$$

*is a subset of universal set P that is BNS.*

Some fundamental operations [12] used for BNSs are stated below:

Let $A_1 = \left\{ \left( \chi, \mathfrak{I}_1^+(\chi), \mathrm{I}_1^+(\chi), f_1^+(\chi), \mathfrak{I}_1^-(\chi), \mathrm{I}_1^-(\chi), f_1^-(\chi) \right) \mid \chi \in P \right\}$ and

$A_2 = \left\{ \left( \chi, \mathfrak{I}_2^+(\chi), \mathrm{I}_2^+(\chi), f_2^+(\chi), \mathfrak{I}_2^-(\chi), \mathrm{I}_2^-(\chi), f_2^-(\chi) \right) \mid \chi \in P \right\}$ be two BNSs.

i. Then, $A_1 \subseteq A_2$ if and only if

$$\mathfrak{I}_1^+(\chi) \le \mathfrak{I}_2^+(\chi), \mathrm{I}_1^+(\chi) \le \mathrm{I}_2^+(\chi), f_1^+(\chi) \ge f_2^+(\chi),$$

and

$$\mathfrak{I}_1^-(\chi) \ge \mathfrak{I}_2^-(\chi), \mathrm{I}_1^-(\chi) \ge \mathrm{I}_2^-(\chi), f_1^-(\chi) \le f_2^-(\chi)$$

ii. $A_1 = A_2$ if and only if

$$\mathfrak{I}_1^+(\chi) = \mathfrak{I}_2^+(\chi), \mathrm{I}_1^+(\chi) = \mathrm{I}_2^+(\chi), f_1^+(\chi) = f_2^+(\chi),$$

and

$$\mathfrak{I}_1^-(\chi) = \mathfrak{I}_2^-(\chi), \mathrm{I}_1^-(\chi) = \mathrm{I}_2^-(\chi), f_1^-(\chi) = f_2^-(\chi)$$

iii. The union is defined as below:

$$(A_1 \cup A_2) = \left\{ \left( \max\left(\mathfrak{I}_1^+(\chi), \mathfrak{I}_2^+(\chi)\right), \frac{\mathrm{I}_1^+(\chi) + \mathrm{I}_2^+(\chi)}{2}, \min\left(f_1^+(\chi), f_2^+(\chi)\right), \min\left(\mathfrak{I}_1^-(\chi), \mathfrak{I}_2^-(\chi)\right), \frac{\mathrm{I}_1^-(\chi) + \mathrm{I}_2^-(\chi)}{2}, \max\left(f_1^-(\chi), f_2^-(\chi)\right) \right) \right\},$$

iv. The intersection is defined as:

$$(A_1 \cap A_2) = \left\{ \left( \min\left(\mathfrak{I}_1^+(\chi), \mathfrak{I}_2^+(\chi)\right), \frac{\mathrm{I}_1^+(\chi) + \mathrm{I}_2^+(\chi)}{2}, \max\left(f_1^+(\chi), f_2^+(\chi)\right), \max\left(\mathfrak{I}_1^-(\chi), \mathfrak{I}_2^-(\chi)\right), \frac{\mathrm{I}_1^-(\chi) + \mathrm{I}_2^-(\chi)}{2}, \min\left(f_1^-(\chi), f_2^-(\chi)\right) \right) \right\},$$

v. Let $A = \left\{ \left( \chi, \mathfrak{I}^+(\chi), \mathrm{I}^+(\chi), f^+(\chi), \mathfrak{I}^-(\chi), \mathrm{I}^-(\chi), f^-(\chi) \right) \mid \chi \in P \right\}$ be BNSs. Then, the complement $A'$ is defined as:

$$\mathfrak{I}_{A'}^+(\chi) = \left\{1^+\right\} - \mathfrak{I}_A^+(\chi), \mathrm{I}_{A'}^+(\chi) = \left\{1^+\right\} - \mathrm{I}_A^+(\chi), f_{A'}^+(\chi) = \left\{1^+\right\} - f_A^+(\chi),$$

and

$$\mathfrak{I}_{A'}^-(\chi) = \left\{1^-\right\} - \mathfrak{I}_A^-(\chi), \mathrm{I}_{A'}^-(\chi) = \left\{1^-\right\} - \mathrm{I}_A^-(\chi), f_{A'}^-(\chi) = \left\{1^-\right\} - f_A^-(\chi),$$

**Definition 10 [25].***Consider two bipolar neutrosophic numbers (BNNs) $u_1 = \left( \mathfrak{I}_1^+, \mathrm{I}_1^+, f_1^+, \mathfrak{I}_1^-, \mathrm{I}_1^-, f_1^- \right)$ and $u_2 = \left( \mathfrak{I}_2^+, \mathrm{I}_2^+, f_2^+, \mathfrak{I}_2^-, \mathrm{I}_2^-, f_2^- \right)$. The following are the basic operations between two BNNs:*

i. $u_1 + u_2 = \left( \mathfrak{I}_1^+ + \mathfrak{I}_2^+ - \mathfrak{I}_1^+ \mathfrak{I}_2^+, \mathrm{I}_1^+ \mathrm{I}_2^+, f_1^+ f_2^+, -\mathfrak{I}_1^- \mathfrak{I}_2^-, -\left(-\mathrm{I}_1^- - \mathrm{I}_2^- - \mathrm{I}_1^- \mathrm{I}_2^-\right), -\left(-f_1^- - f_2^- - f_1^- f_2^-\right) \right);$

ii. $u_1 \cdot u_2 = \left( \mathfrak{I}_1^+ \mathfrak{I}_2^+, \mathrm{I}_1^+ + \mathrm{I}_2^+ - \mathrm{I}_1^+ \mathrm{I}_2^+, f_1^+ + f_2^+ - f_1^+ f_2^+, -\left(-\mathfrak{I}_1^- - \mathfrak{I}_2^- - \mathfrak{I}_1^- \mathfrak{I}_2^-\right), -\mathrm{I}_1^- \mathrm{I}_2^-, -f_1^- f_2^- \right);$

iii. $\lambda(u_1) = \left( 1 - \left(1 - \mathfrak{I}_1^+\right)^\lambda, \left(\mathrm{I}_1^+\right)^\lambda, \left(f_1^+\right)^\lambda, -\left(-\mathfrak{I}_1^-\right)^\lambda, -\left(-\mathrm{I}_1^-\right)^\lambda, -\left(1 - \left(1 - \left(-f_1^-\right)\right)^\lambda\right) \right);$

$$\text{iv.}\,\left(u_1\right)^{\lambda}=\left(\left(\mathfrak{I}_1^{+}\right)^{\lambda},1-\left(1-\mathrm{I}_1^{+}\right)^{\lambda},1-\left(1-f_1^{+}\right)^{\lambda},-\left(1-\left(1-\left(-\mathfrak{I}_1^{-}\right)\right)^{\lambda}\right),-\left(-\mathrm{I}_1^{-}\right)^{\lambda},-\left(-f_1^{-}\right)^{\lambda}\right);$$

*where* $\lambda>0.$

**Definition 11 [25].** *The score function for a bipolar neutrosophic number* $u=\left(\mathfrak{I}^{+},\mathrm{I}^{+},f^{+},\mathfrak{I}^{-},\mathrm{I}^{-},f^{-}\right)$ *denoted by* $S(u)$ *is:*

$$S(u)=\frac{1}{6}\left(\mathfrak{I}^{+}+1-\mathrm{I}^{+}+1-f^{+}+1+\mathfrak{I}^{-}-\mathrm{I}^{-}-f^{-}\right)$$

**Definition 12 [25].** *The accuracy function defined for a BNN* $u=\left(\mathfrak{I}^{+},\mathrm{I}^{+},f^{+},\mathfrak{I}^{-},\mathrm{I}^{-},f^{-}\right)$ *is* $a\left(u\right)$, *as below:*

$$a\left(u\right)=\mathfrak{I}^{+}-f^{+}+\mathfrak{I}^{-}-f^{-}$$

**Definition 13 [25].** *Consider* $u=\left(\mathfrak{I}^{+},\mathrm{I}^{+},f^{+},\mathfrak{I}^{-},\mathrm{I}^{-},f^{-}\right)$ *to represent the neutrosophic number. Then, thecertainty function is:*

$$c\left(u\right)=\mathfrak{I}^{+}-f^{-}$$

**Definition 14 [25].** *Consider* $u_1=\left(\mathfrak{I}_1^{+},\mathrm{I}_1^{+},f_1^{+},\mathfrak{I}_1^{-},\mathrm{I}_1^{-},f_1^{-}\right)$ *and* $u_2=\left(\mathfrak{I}_2^{+},\mathrm{I}_2^{+},f_2^{+},\mathfrak{I}_2^{-},\mathrm{I}_2^{-},f_2^{-}\right)$ *to represent two BNNs. Then, acomparison method among BNNs is:*

i.    *In condition* $S(u_1)\succ S(u_2)$ *then* $u_1$ *is greater than* $u_2$, *denoted by* $u_1\succ u_2$

ii.    *In condition* $S(u_1)=S(u_2)$ *and* $a(u_1)\succ a(u_2)$ *then* $u_1$ *is greater than* $u_2$, *denoted by* $u_1\succ u_2$;

iii.    *In condition* $S(u_1)=S(u_2)$, $a(u_1)=a(u_2)$ *and* $c(u_1)\succ c(u_2)$ *in that case,* $u_1$ *is superior to* $u_2$, *denote by* $u_1\succ u_2$;

iv.    *In condition* $S(u_1)=S(u_2)$, $a(u_1)=a(u_2)$ *and* $c(u_1)=c(u_2)$ *in that case,* $u_1$ *is equal to* $u_2$, *denoted by* $u_1=u_2.$

**Definition 15.** *Consider* $u=\left(\mathfrak{I}^{+},\mathrm{I}^{+},f^{+},\mathfrak{I}^{-},\mathrm{I}^{-},f^{-}\right)$, $u_1=\left(\mathfrak{I}_1^{+},\mathrm{I}_1^{+},f_1^{+},\mathfrak{I}_1^{-},\mathrm{I}_1^{-},f_1^{-}\right)$ *and* $u_2=\left(\mathfrak{I}_2^{+},\mathrm{I}_2^{+},f_2^{+},\mathfrak{I}_2^{-},\mathrm{I}_2^{-},f_2^{-}\right)$ *to be BNNs, and* $\lambda\succ0$ *any real value. Then, the fundamental Einstein operations of BNNs are:*

$$u_1\oplus u_2=\left(\frac{\mathfrak{I}_1^{+}+\mathfrak{I}_2^{+}}{1+\mathfrak{I}_1^{+}\mathfrak{I}_2^{+}},\frac{\mathrm{I}_1^{+}\mathrm{I}_2^{+}}{1+\left(1-\mathrm{I}_1^{+}\right)\left(1-\mathrm{I}_2^{+}\right)},\frac{f_1^{+}f_2^{+}}{1+\left(1-f_1^{+}\right)\left(1-f_2^{+}\right)},\frac{-\mathfrak{I}_1^{-}\mathfrak{I}_2^{-}}{1+\left(1+\mathfrak{I}_1^{-}\right)\left(1+\mathfrak{I}_2^{-}\right)},\frac{-\left(-\mathrm{I}_1^{-}-\mathrm{I}_2^{-}\right)}{1+\mathrm{I}_1^{-}\mathrm{I}_2^{-}},\frac{-\left(-f_1^{-}-f_2^{-}\right)}{1+f_1^{-}f_2^{-}}\right)$$

$$u_1 \otimes u_2 = \left( \frac{\mathfrak{I}_1^+ \mathfrak{I}_2^+}{1+\left(1-\mathfrak{I}_1^+\right)\left(1-\mathfrak{I}_2^+\right)}, \frac{I_1^+ + I_2^+}{1+I_1^+ I_2^+}, \frac{f_1^+ + f_2^+}{1+f_1^+ f_2^+}, \frac{-\left(-\mathfrak{I}_1^- - \mathfrak{I}_2^-\right)}{1+\mathfrak{I}_1^- \mathfrak{I}_2^-}, \frac{-I_1^- I_2^-}{1+\left(1+I_1^-\right)\left(1+I_2^-\right)}, \frac{-f_1^- f_2^-}{1+\left(1+f_1^-\right)\left(1+f_2^-\right)} \right)$$

$$\lambda(u) = \left( \frac{\left(1+\mathfrak{I}^+\right)^\lambda - \left(1-\mathfrak{I}^+\right)^\lambda}{\left(1+\mathfrak{I}^+\right)^\lambda + \left(1-\mathfrak{I}^+\right)^\lambda}, \frac{2\left(I^+\right)^\lambda}{\left(2-I^+\right)^\lambda + \left(I^+\right)^\lambda}, \frac{2\left(f^+\right)^\lambda}{\left(2-f^+\right)^\lambda + \left(f^+\right)^\lambda}, \frac{-2\left|\mathfrak{I}^-\right|^\lambda}{\left(2+\mathfrak{I}^-\right)^\lambda + \left|\mathfrak{I}^-\right|^\lambda}, \frac{-2\left|I^-\right|^\lambda}{\left(2+I^-\right)^\lambda + \left|I^-\right|^\lambda}, \frac{-\left(\left(1+\left|f^-\right|\right)^\lambda - \left(1+f^-\right)^\lambda\right)}{\left(1+\left|f^-\right|\right)^\lambda + \left(1+f^-\right)^\lambda} \right)$$

$$(u)^\lambda = \left( \frac{2\left(\mathfrak{I}^+\right)^\lambda}{\left(2-\mathfrak{I}^+\right)^\lambda + \left(\mathfrak{I}^+\right)^\lambda}, \frac{\left(1+I^+\right)^\lambda - \left(1-I^+\right)^\lambda}{\left(1+I^+\right)^\lambda + \left(1-I^+\right)^\lambda}, \frac{\left(1+f^+\right)^\lambda - \left(1-f^+\right)^\lambda}{\left(1+f^+\right)^\lambda + \left(1-f^+\right)^\lambda}, \frac{-\left(\left(1+\left|\mathfrak{I}^-\right|\right)^\lambda - \left(1+\mathfrak{I}^-\right)^\lambda\right)}{\left(1+\left|\mathfrak{I}^-\right|\right)^\lambda + \left(1+\mathfrak{I}^-\right)^\lambda}, \frac{-2\left|I^-\right|^\lambda}{\left(2+I^-\right)^\lambda + \left|I^-\right|^\lambda}, \frac{-2\left|f^-\right|^\lambda}{\left(2+f^-\right)^\lambda + \left|f^-\right|^\lambda} \right)$$

### 3. Bipolar Neutrosophic Einstein Average AOs

Here, we suggest a number of fundamental properties for bipolar Einsteinaverage AOs in the current section of the article, such as BNEWA, BNEOWA and BNEHA.

*3.1. Bipolar NeutrosophicEinstein Weighted-Average Aggregation Operators*

Consider $u_\ell = \left( \mathfrak{I}_\ell^+, I_\ell^+, f_\ell^+, \mathfrak{I}_\ell^-, I_\ell^-, f_\ell^- \right)$ to represent a family of BNNs. Here, $\ell \in \{1,2,3,...,n\}$

**Definition 16.** *The (BNEWA) aggregation operator is defined as follows:*

$$BNEWA_v\left(u_1, u_2, ....., u_n\right) = \overset{n}{\underset{\ell=1}{\oplus}}\left(v_\ell u_\ell\right) = v_1 u_1 \oplus v_2 u_2 \oplus ......... \oplus v_n u_n \tag{1}$$

*where* $v = \left(v_1, v_2, ..., v_n\right)^T$ *represents weighted vectors of* $u_\ell$, *that is,* $v_\ell > 0$ *and* $\sum_{\ell=1}^{n} v_\ell = 1.$

**Theorem1.** *The (BNEWA) operator gives in return a (BNV) by*

$$BNEWA_v\left(u_1, u_2, ....., u_n\right) =$$
$$\left( \frac{\mathcal{B}_1 - \mathcal{B}_2}{\mathcal{B}_1 + \mathcal{B}_2}, \frac{2\zeta_1}{\zeta_2 + \zeta_1}, \frac{2f_1}{f_2 + f_1}, \frac{-2\mathcal{B}_3}{\mathcal{B}_4 + \mathcal{B}_3}, -\frac{\zeta_3 - \zeta_4}{\zeta_3 + \zeta_4}, -\frac{f_3 - f_4}{f_3 + f_4} \right) \tag{2}$$

*where*

$$\mathcal{B}_1 = \prod_{\ell=1}^{n}\left(1+\mathfrak{I}_\ell^+\right)^{v_\ell}, \quad \mathcal{B}_2 = \prod_{\ell=1}^{n}\left(1-\mathfrak{I}_\ell^+\right)^{v_\ell}, \quad \mathcal{B}_3 = \prod_{\ell=1}^{n}\left|\mathfrak{I}_\ell^-\right|^{v_\ell}, \quad \mathcal{B}_4 = \prod_{\ell=1}^{n}\left(2+\mathfrak{I}_\ell^-\right)^{v_\ell},$$

$$\zeta_1 = \prod_{\ell=1}^{n}\left(I_\ell^+\right)^{v_\ell}, \quad \zeta_2 = \prod_{\ell=1}^{n}\left(2-I_\ell^+\right)^{v_\ell}, \quad \zeta_3 = \prod_{\ell=1}^{n}\left(1+\left|I_\ell^-\right|\right)^{v_\ell}, \quad \zeta_4 = \prod_{\ell=1}^{n}\left(1+I_\ell^-\right)^{v_\ell},$$

$$f_1 = \prod_{\ell=1}^{n}\left(f_\ell^+\right)^{v_\ell}, \quad f_2 = \prod_{\ell=1}^{n}\left(2-f_\ell^+\right)^{v_\ell}, \quad f_3 = \prod_{\ell=1}^{n}\left(1+\left|f_\ell^-\right|\right)^{v_\ell}, \quad f_4 = \prod_{\ell=1}^{n}\left(1+f_\ell^-\right)^{v_\ell}.$$

*where* $v = \left(v_1, v_2, ..., v_n\right)^T$ *is the weighted vector of* $u_\ell$, *such that* $v_\ell > 0$ *and* $\sum_{\ell=1}^{n} v_\ell = 1.$

**Proof**. Now, by mathematical induction:

For $n = 2$,

$$v_1 u_1 = \begin{pmatrix} \dfrac{\left(1+\Im_1^+\right)^{v_1}-\left(1-\Im_1^+\right)^{v_1}}{\left(1+\Im_1^+\right)^{v_1}+\left(1-\Im_1^+\right)^{v_1}}, \dfrac{2\left(I_1^+\right)^{v_1}}{\left(2-I_1^+\right)^{v_1}+\left(I_1^+\right)^{v_1}}, \dfrac{2\left(f_1^+\right)^{v_1}}{\left(2-f_1^+\right)^{v_1}+\left(f_1^+\right)^{v_1}}, \\[3ex] \dfrac{-2\left|\Im_1^-\right|^{v_1}}{\left(2+\Im_1^-\right)^{v_1}+\left|\Im_1^-\right|^{v_1}}, \dfrac{-2\left|I_1^-\right|^{v_1}}{\left(2+I_1^-\right)^{v_1}+\left|I_1^-\right|^{v_1}}, \dfrac{-\left(\left(1+\left|f_1^-\right|\right)^{v_1}-\left(1+f_1^-\right)^{v_1}\right)}{\left(1+\left|f_1^-\right|\right)^{v_1}+\left(1+f_1^-\right)^{v_1}} \end{pmatrix}$$

and

$$v_2 u_2 = \begin{pmatrix} \dfrac{\left(1+\Im_2^+\right)^{v_2}-\left(1-\Im_2^+\right)^{v_2}}{\left(1+\Im_2^+\right)^{v_2}+\left(1-\Im_2^+\right)^{v_2}}, \dfrac{2\left(I_2^+\right)^{v_2}}{\left(2-I_2^+\right)^{v_2}+\left(I_2^+\right)^{v_2}}, \dfrac{2\left(f_2^+\right)^{v_2}}{\left(2-f_2^+\right)^{v_2}+\left(f_2^+\right)^{v_2}}, \\[3ex] \dfrac{-2\left|\Im_2^-\right|^{v_2}}{\left(2+\Im_2^-\right)^{v_2}+\left|\Im_2^-\right|^{v_2}}, \dfrac{-2\left|I_2^-\right|^{v_2}}{\left(2+I_2^-\right)^{v_2}+\left|I_2^-\right|^{v_2}}, \dfrac{-\left(\left(1+\left|f_2^-\right|\right)^{v_2}-\left(1+f_2^-\right)^{v_2}\right)}{\left(1+\left|f_2^-\right|\right)^{v_2}+\left(1+f_2^-\right)^{v_2}} \end{pmatrix}$$

and for

$$v_1 u_1 = \begin{pmatrix} \dfrac{\left(1+\Im_1^+\right)-\left(1-\Im_1^+\right)}{\left(1+\Im_1^+\right)+\left(1-\Im_1^+\right)}, \dfrac{2\left(I_1^+\right)}{\left(2-I_1^+\right)+\left(I_1^+\right)}, \dfrac{2\left(f_1^+\right)}{\left(2-f_1^+\right)+\left(f_1^+\right)}, \\[3ex] \dfrac{-2\left|\Im_1^-\right|}{\left(2+\Im_1^-\right)+\left|\Im_1^-\right|}, -\dfrac{\left(1+\left|I_1^-\right|\right)-\left(1+I_1^-\right)}{\left(1+\left|I_1^-\right|\right)+\left(1+I_1^-\right)}, -\dfrac{\left(1+\left|f_1^-\right|\right)-\left(1+f_1^-\right)}{\left(1+\left|f_1^-\right|\right)+\left(1+f_1^-\right)} \end{pmatrix}, \qquad (3)$$

thus satisfying $n = 1$.

We put $n = r$ into Equation (3)

$$BNEWA_v\left(u_1, u_2, \ldots, u_r\right) =$$

$$\begin{pmatrix} \dfrac{\prod\limits_{\ell=1}^{r}\left(1+\Im_\ell^+\right)^{v_\ell}-\prod\limits_{\ell=1}^{r}\left(1-\Im_\ell^+\right)^{v_\ell}}{\prod\limits_{\ell=1}^{r}\left(1+\Im_\ell^+\right)^{v_\ell}+\prod\limits_{\ell=1}^{r}\left(1-\Im_\ell^+\right)^{v_\ell}}, \dfrac{2\prod\limits_{\ell=1}^{r}\left(I_\ell^+\right)^{v_\ell}}{\prod\limits_{\ell=1}^{r}\left(2-I_\ell^+\right)^{v_\ell}+\prod\limits_{\ell=1}^{r}\left(I_\ell^+\right)^{v_\ell}}, \dfrac{2\prod\limits_{\ell=1}^{r}\left(f_\ell^+\right)^{v_\ell}}{\prod\limits_{\ell=1}^{r}\left(2-f_\ell^+\right)^{v_\ell}+\prod\limits_{\ell=1}^{r}\left(f_\ell^+\right)^{v_\ell}}, \\[4ex] \dfrac{-2\prod\limits_{\ell=1}^{r}\left|\Im_\ell^-\right|^{v_\ell}}{\prod\limits_{\ell=1}^{r}\left(2+\Im_\ell^-\right)^{v_\ell}+\prod\limits_{\ell=1}^{r}\left|\Im_\ell^-\right|^{v_\ell}}, \dfrac{-2\prod\limits_{\ell=1}^{r}\left|I_\ell^-\right|^{v_\ell}}{\prod\limits_{\ell=1}^{r}\left(2+I_\ell^-\right)^{v_\ell}+\prod\limits_{\ell=1}^{r}\left|I_\ell^-\right|^{v_\ell}}, -\dfrac{\prod\limits_{\ell=1}^{k}\left(1+\left|f_\ell^-\right|\right)^{v_\ell}-\prod\limits_{\ell=1}^{r}\left(1+f_\ell^-\right)^{v_\ell}}{\prod\limits_{\ell=1}^{r}\left(1+\left|f_\ell^-\right|\right)^{v_\ell}+\prod\limits_{\ell=1}^{r}\left(1+f_\ell^-\right)^{v_\ell}} \end{pmatrix},$$

Assume it is also satisfied for $n = r$.

Now, for $n = r+1,$

$$BNEWA_\nu\left(u_1,u_2,.....,u_r\right)=$$

$$\left(\begin{array}{c}\dfrac{\prod\limits_{\ell=1}^{r}\left(1+\mathfrak{I}_\ell^+\right)^{\nu_\ell}-\prod\limits_{\ell=1}^{r}\left(1-\mathfrak{I}_\ell^+\right)^{\nu_\ell}}{\prod\limits_{\ell=1}^{r}\left(1+\mathfrak{I}_\ell^+\right)^{\nu_\ell}+\prod\limits_{\ell=1}^{r}\left(1-\mathfrak{I}_\ell^+\right)^{\nu_\ell}},\dfrac{2\prod\limits_{\ell=1}^{r}\left(I_\ell^+\right)^{\nu_\ell}}{\prod\limits_{\ell=1}^{r}\left(2-I_\ell^+\right)^{\nu_\ell}+\prod\limits_{\ell=1}^{r}\left(I_\ell^+\right)^{\nu_\ell}},\dfrac{2\prod\limits_{\ell=1}^{r}\left(f_\ell^+\right)^{\nu_\ell}}{\prod\limits_{\ell=1}^{r}\left(2-f_\ell^+\right)^{\nu_\ell}+\prod\limits_{\ell=1}^{r}\left(f_\ell^+\right)^{\nu_\ell}},\\[4mm]\dfrac{-2\prod\limits_{\ell=1}^{r}\left|\mathfrak{I}_\ell^-\right|^{\nu_\ell}}{\prod\limits_{\ell=1}^{r}\left(2+\mathfrak{I}_\ell^-\right)^{\nu_\ell}+\prod\limits_{\ell=1}^{r}\left|\mathfrak{I}_\ell^-\right|^{\nu_\ell}},\dfrac{-2\prod\limits_{\ell=1}^{r}\left|I_\ell^-\right|^{\nu_\ell}}{\prod\limits_{\ell=1}^{r}\left(2+I_\ell^-\right)^{\nu_\ell}+\prod\limits_{\ell=1}^{r}\left|I_\ell^-\right|^{\nu_\ell}},-\dfrac{\prod\limits_{\ell=1}^{r}\left(1+\left|f_\ell^-\right|\right)^{\nu_\ell}-\prod\limits_{\ell=1}^{r}\left(1+f_\ell^-\right)^{\nu_\ell}}{\prod\limits_{\ell=1}^{r}\left(1+\left|f_\ell^-\right|\right)^{\nu_\ell}+\prod\limits_{\ell=1}^{r}\left(1+f_\ell^-\right)^{\nu_\ell}}\end{array}\right)$$

$$\oplus\left(\begin{array}{c}\dfrac{\left(1+\mathfrak{I}_{r+1}^+\right)^{\nu_{r+1}}-\left(1-\mathfrak{I}_{r+1}^+\right)^{\nu_{r+1}}}{\left(1+\mathfrak{I}_{r+1}^+\right)^{\nu_{r+1}}+\left(1-\mathfrak{I}_{r+1}^+\right)^{\nu_{r+1}}},\dfrac{2\left(I_{r+1}^+\right)^{\nu_{r+1}}}{\left(2-I_{r+1}^+\right)^{\nu_{r+1}}+\left(I_{r+1}^+\right)^{\nu_{r+1}}},\dfrac{2\left(f_{r+1}^+\right)^{\nu_{r+1}}}{\left(2-f_{r+1}^+\right)^{\nu_{r+1}}+\left(f_{r+1}^+\right)^{\nu_{r+1}}},\\[4mm]\dfrac{-2\left|\mathfrak{I}_{r+1}^-\right|^{\nu_{r+1}}}{\left(2+\mathfrak{I}_{r+1}^-\right)^{\nu_{r+1}}+\left|\mathfrak{I}_{r+1}^-\right|^{\nu_{r+1}}},\dfrac{-2\left|I_{r+1}^-\right|^{\nu_{r+1}}}{\left(2+I_{r+1}^-\right)^{\nu_{r+1}}+\left|I_{r+1}^-\right|^{\nu_{r+1}}},-\dfrac{\left(1+\left|f_{r+1}^-\right|\right)^{\nu_{r+1}}-\left(1+f_{r+1}^-\right)^{\nu_{r+1}}}{\left(1+\left|f_{r+1}^-\right|\right)^{\nu_{r+1}}+\left(1+f_{r+1}^-\right)^{\nu_{r+1}}}\end{array}\right)$$

$$=\left(\begin{array}{c}\dfrac{\prod\limits_{\ell=1}^{r+1}\left(1+\mathfrak{I}_\ell^+\right)^{\nu_\ell}-\prod\limits_{\ell=1}^{r+1}\left(1-\mathfrak{I}_\ell^+\right)^{\nu_\ell}}{\prod\limits_{\ell=1}^{r+1}\left(1+\mathfrak{I}_\ell^+\right)^{\nu_\ell}+\prod\limits_{\ell=1}^{r+1}\left(1-\mathfrak{I}_\ell^+\right)^{\nu_\ell}},\dfrac{2\prod\limits_{\ell=1}^{r+1}\left(I_\ell^+\right)^{\nu_\ell}}{\prod\limits_{\ell=1}^{r+1}\left(2-I_\ell^+\right)^{\nu_\ell}+\prod\limits_{\ell=1}^{r+1}\left(I_\ell^+\right)^{\nu_\ell}},\dfrac{2\prod\limits_{\ell=1}^{r+1}\left(f_\ell^+\right)^{\nu_\ell}}{\prod\limits_{\ell=1}^{r+1}\left(2-f_\ell^+\right)^{\nu_\ell}+\prod\limits_{\ell=1}^{r+1}\left(f_\ell^+\right)^{\nu_\ell}},\\[4mm]\dfrac{-2\prod\limits_{\ell=1}^{r+1}\left|\mathfrak{I}_\ell^-\right|^{\nu_\ell}}{\prod\limits_{\ell=1}^{r+1}\left(2+\mathfrak{I}_\ell^-\right)^{\nu_\ell}+\prod\limits_{\ell=1}^{r+1}\left|\mathfrak{I}_\ell^-\right|^{\nu_\ell}},\dfrac{\prod\limits_{\ell=1}^{r+1}\left(1+\left|I_\ell^-\right|\right)^{\nu_\ell}-\prod\limits_{\ell=1}^{r+1}\left(1+I_\ell^-\right)^{\nu_\ell}}{\prod\limits_{\ell=1}^{r+1}\left(1+\left|I_\ell^-\right|\right)^{\nu_\ell}+\prod\limits_{\ell=1}^{r+1}\left(1+I_\ell^-\right)^{\nu_\ell}},-\dfrac{\prod\limits_{\ell=1}^{r+1}\left(1+\left|f_\ell^-\right|\right)^{\nu_\ell}-\prod\limits_{\ell=1}^{r+1}\left(1+f_\ell^-\right)^{\nu_\ell}}{\prod\limits_{\ell=1}^{r+1}\left(1+\left|f_\ell^-\right|\right)^{\nu_\ell}+\prod\limits_{\ell=1}^{r+1}\left(1+f_\ell^-\right)^{\nu_\ell}}\end{array}\right)$$

Hence, Equation (3) is satisfied for all values of $n=r+1$.

Hence, the theorem is proven. $\square$

**Theorem 2.** *(Idempotency) Let* $u_\ell=\left(\mathfrak{I}_\ell^+,I_\ell^+,f_\ell^+,\mathfrak{I}_\ell^-,I_\ell^-,f_\ell^-\right)$, *where* $\ell\in\left\{1,2,...,n-1,n\right\}$ *represents a collection of equal BNNs, that is,* $u_\ell=u$*. Then*

$$BNEWA_\nu\left(u_1,u_2,.....,u_n\right)=u$$

**Proof**. Since

$$BNEWA_\nu\left(u_1,u_2,.....,u_n\right)$$
$$=\overset{n}{\underset{\ell=1}{\oplus}}\left(\nu_\ell u_\ell\right)=\overset{n}{\underset{\ell=1}{\oplus}}\left(\nu_\ell\right)u=\left(u\right)\sum_{\ell=1}^{n}\nu_\ell=u$$

the proof is complete. $\square$

**Theorem 3.** *(Bounded) Suppose that* $u^+=\max\limits_\ell u_\ell$ *and* $u^-=\min\limits_\ell u_\ell$ *represent the minimum and maximum BNN, respectively*

$$u^- \leq BNEWA_v\left(u_1, u_2, \ldots, u_n\right) \leq u^+$$

**Proof.** Let

$$BNEWA_v\left(u_1, u_2, \ldots, u_n\right) = u\left(\mathfrak{I}^+, \mathrm{I}^+, f^+, \mathfrak{I}^-, \mathrm{I}^-, f^-\right)$$

Then,

$$s\left(u^-\right) \leq s\left(BNEWA \text{ operator}\right)$$

$$s\left(u^-\right) \leq s\left(BNEWA \text{ operator}\right)$$

$$s\left(u^+\right) \geq s\left(BNEWA \text{ operator}\right)$$

Combining both equations, we have

$$u^- \leq BNEWA_v\left(u_1, u_2, \ldots, u_n\right) \leq u^+$$

Thus, the proof is complete. □

**Theorem 4.** *(Monotonicity) Suppose* $u_\ell = \left(\mathfrak{I}_\ell^+, \mathrm{I}_\ell^+, f_\ell^+, \mathfrak{I}_\ell^-, \mathrm{I}_\ell^-, f_\ell^-\right)$ *and*
$u_\ell' = \left(\mathfrak{I}_\ell'^+, \mathrm{I}_\ell'^+, f_\ell'^+, \mathfrak{I}_\ell'^-, \mathrm{I}_\ell'^-, f_\ell'^-\right)$, *where* $\ell \in \{1, 2, \ldots, n-1, n\}$ *represents BNNs. If* $u_\ell \leq u_\ell'$,
*then*

$$BNEWA_v\left(u_1, u_2, \ldots, u_n\right) \leq BNEWA_v\left(u_1', u_2', \ldots, u_n'\right)$$

**Proof.** Assume that

$$BNEWA_v\left(u_1, u_2, \ldots, u_n\right) = \overset{n}{\underset{\ell=1}{\oplus}}\left(v_\ell u_\ell\right)$$

and

$$BNEWA_v\left(u_1', u_2', \ldots, u_n'\right) = \overset{n}{\underset{\ell=1}{\oplus}}\left(v_\ell u_\ell'\right)$$

Since $u_\ell \leq u_\ell'$, then

$$\overset{n}{\underset{\ell=1}{\oplus}}\left(v_\ell u_\ell\right) \leq \overset{n}{\underset{\ell=1}{\oplus}}\left(v_\ell u_\ell'\right)$$

Thus, the proof is complete. □

*3.2. BN Einstein OrderedWeighted Average Aggregation Operators*

**Definition 17.** *The (BNEOWA)BN Einstein ordered weighted-average AOis stated as:*

$$BNEOWA_v\left(u_1, u_2, \ldots, u_n\right) = \overset{n}{\underset{\ell=1}{\oplus}}\left(v_\ell u_{\rho(\ell)}\right) = v_1 u_{\rho(1)} \oplus v_2 u_{\rho(2)} \oplus v_3 u_{\sigma(3)} \oplus \ldots \oplus v_n u_{\rho(n)}, \tag{4}$$

where $\left(\rho(1),\rho(2),\rho(3),...,\rho(n)\right)$ represents a permutation with $u_{\rho(\ell-1)}\geq u_{\rho(\ell)},\forall\,\ell\in Z$ , $Z=\{1,2,....,n-1,n\}$ and $v=\left(v_1,v_2,...,v_n\right)^T$ representing associated weighting vectors for $u_\ell$ with $v_\ell>0$, $\sum\limits_{\ell=1}^{n}v_\ell=1$.

**Theorem 5.** *The (BNEOWA) operator gives, in return, a BNV, by*

$$BNEOWA_v\left(u_1,u_2,.....,u_n\right)=$$

$$\left(\frac{\mathcal{B}_{\partial 1}-\mathcal{B}_{\partial 2}}{\mathcal{B}_{\partial 1}+\mathcal{B}_{\partial 2}},\frac{2\zeta_{\partial 1}}{\zeta_{\partial 2}+\zeta_{\partial 1}},\frac{2f_{\partial 1}}{f_{\partial 2}+f_{\partial 1}},\frac{-2\mathcal{B}_{\partial 3}}{\mathcal{B}_{\partial 4}+\mathcal{B}_{\partial 3}},-\frac{\zeta_{\partial 3}-\zeta_{\partial 4}}{\zeta_{\partial 3}+\zeta_{\partial 4}},-\frac{f_{\partial 3}-f_{\partial 4}}{f_{\partial 3}+f_{\partial 4}}\right) \tag{5}$$

*where,*

$$\mathcal{B}_{\partial 1}=\prod_{\ell=1}^{n}\left(1+\mathfrak{I}_{\rho(\ell)}^{+}\right)^{v_\ell},\quad \mathcal{B}_{\partial 2}=\prod_{\ell=1}^{n}\left(1-\mathfrak{I}_{\rho(\ell)}^{+}\right)^{v_\ell},\quad \mathcal{B}_{\partial 3}=\prod_{\ell=1}^{n}\left|\mathfrak{I}_{\rho(\ell)}^{-}\right|^{v_\ell},\quad \mathcal{B}_{\partial 4}=\prod_{\ell=1}^{n}\left(2+\mathfrak{I}_{\rho(\ell)}^{-}\right)^{v_\ell},$$

$$\zeta_{\partial 1}=\prod_{\ell=1}^{n}\left(I_{\rho(\ell)}^{+}\right)^{v_\ell},\quad \zeta_{\partial 2}=\prod_{\ell=1}^{n}\left(2-I_{\rho(\ell)}^{+}\right)^{v_\ell},\quad \zeta_{\partial 3}=\prod_{\ell=1}^{n}\left(1+\left|I_{\rho(\ell)}^{-}\right|\right)^{v_\ell},\quad \zeta_{\partial 4}=\prod_{\ell=1}^{n}\left(1+I_{\rho(\ell)}^{-}\right)^{v_\ell},$$

$$f_{\partial 1}=\prod_{\ell=1}^{n}\left(f_{\rho(\ell)}^{+}\right)^{v_\ell},\quad f_{\partial 2}=\prod_{\ell=1}^{n}\left(2-f_{\rho(\ell)}^{+}\right)^{v_\ell},\quad f_{\partial 3}=\prod_{\ell=1}^{n}\left(1+\left|f_{\rho(\ell)}^{-}\right|\right)^{v_\ell},\quad f_{\partial 4}=\prod_{\ell=1}^{n}\left(1+f_{\rho(\ell)}^{-}\right)^{v_\ell}.$$

where $\left(\rho(1),\rho(2),.........,\rho(n)\right)$ represents a permutation with $u_{\rho(\ell-1)}\geq u_{\rho(\ell)}$, $\forall\,\ell\in Z$ , $Z=\{1,2,...,n-1,n\}$ and $v=\left(v_1,v_2,...,v_n\right)^T$ representing associated weighting vectors for $u_\ell$ with $v_\ell>0$ and $\sum\limits_{\ell=1}^{n}v_\ell=1$.

**Proof.** The prooffollowsfrom Theorem 1. □

**Theorem 6.** *(Commutativity) Let* $u_\ell=\left(\mathfrak{I}_\ell^{+},I_\ell^{+},f_\ell^{+},\mathfrak{I}_\ell^{-},I_\ell^{-},f_\ell^{-}\right)$, *as well as* $u_\ell'=\left(\mathfrak{I}_\ell'^{+},I_\ell'^{+},f_\ell'^{+},\mathfrak{I}_\ell'^{-},I_\ell'^{-},f_\ell'^{-}\right)$, *where* $\ell\in Z$ *represents two BNNs.*

$$BNEOWA_\varpi\left(u_1,u_2,...,u_n\right)=BNEOWA_\varpi\left(u_1',u_2',...,u_n'\right)$$

*where* $\left(u_1,u_2,...,u_n\right)$ *is any permutation of* $\left(u_1',u_2',...,u_n'\right)$.

**Proof.** Assume that

$$BNEOWA_v\left(u_1,u_2,...,u_n\right)=\bigoplus_{\ell=1}^{n}\left(v_\ell u_{\rho(\ell)}\right)$$

and

$$BNEOWA_v\left(u_1',u_2',...,u_n'\right)=\bigoplus_{\ell=1}^{n}\left(v_\ell u_{\rho(\ell)}'\right)$$

Since $\left(u_1,u_2,...,u_n\right)$ is any permutation of $\left(u_1',u_2',...,u_n'\right)$, then

$$\overset{n}{\underset{\ell=1}{\oplus}}\left(\nu_\ell u_{\rho(\ell)}\right) = \overset{n}{\underset{\ell=1}{\oplus}}\left(\nu_\ell u'_{\rho(\ell)}\right)$$

The proof is complete.□

**Theorem 7.** *(Idempotency) Let* $u_\ell = \left(\mathfrak{I}_\ell^+, I_\ell^+, f_\ell^+, \mathfrak{I}_\ell^-, I_\ell^-, f_\ell^-\right)$, *where* $\ell \in Z$ . $Z = \left\{ 1,2,...,n-1,n\right\}$ *is a collection of all equal BNNs, i.e.,* $u_\ell = u$ :

$$BNEOWA_\nu\left(u_1, u_2, u_3,...,u_n\right) = u$$

**Proof.** The proof follows from Theorem 2.□

**Theorem 8.** *(Bounded) Consider* $u^- = \underset{\ell}{\min}\, u_\ell$, $u^+ = \underset{\ell}{\max}\, u_\ell$ *to represent the minimum and maximum BNN, respectively*

$$u^- \le BNEOWA_\nu\left(u_1, u_2,.....,u_n\right) \le u^+$$

**Proof.** The proof follows from Theorem 3.□

**Theorem 9.** *(Monotonicity) Let* $u_\ell = \left(\mathfrak{I}_\ell^+, I_\ell^+, f_\ell^+, \mathfrak{I}_\ell^-, I_\ell^-, f_\ell^-\right)$, *and* $u'_\ell = \left(\mathfrak{I}_\ell'^+, I_\ell'^+, f_\ell'^+, \mathfrak{I}_\ell'^-, I_\ell'^-, f_\ell'^-\right)$, *where* $\ell \in Z$ *represents BNNs. If* $u_\ell \le u'_\ell$,

$$BNEOWA_\varpi\left(u_1, u_2,.....,u_n\right) \le BNEOWA_\varpi\left(u'_1, u'_2,.....,u'_n\right)$$

**Proof.** The proof follows from Theorem 4.□

*3.3. BN Einstein Hybrid Average Aggregation Operators*

**Definition 18.** *The (BNEHA) BN Einstein hybrid average AO is stated as:*

$$BNEHA_{w,\nu}\left(u_1, u_2,......,u_n\right) = \overset{n}{\underset{\ell=1}{\oplus}}\left(\nu_\ell \dot{u}_{\rho(\ell)}\right) = \nu_1 \dot{u}_{\rho(1)} \oplus \nu_2 \dot{u}_{\rho(2)} \oplus \nu_3 \dot{u}_{\rho(3)} \oplus ...........\oplus \nu_n \dot{u}_{\rho(n)}, \qquad (6)$$

*where* $w = \left(w_1, w_2, w_3,..., w_{n-1}, w_n\right)$ *is a weighting vector of* $u_\ell$, *such that* $w_\ell \in \left[0,\ 1\right]$, $\displaystyle\sum_{\ell=1}^{n} w_\ell = 1$ *and* $\dot{u}_{\rho(\ell)}$ *are the* $\ell$ *-th biggest component of the BN argument* $\dot{u}_\ell\left(\dot{u}_\ell = \left(n\nu_\ell\right)u_\ell, \ell = 1,2,...,n\right)$, *and* $\nu = \left(\nu_1, \nu_2,..., \nu_n\right)$ *is the weighting vector of BN argument* $u_\ell$, *along with* $\nu_\ell \in \left[0,1\right]$, $\displaystyle\sum_{\ell=1}^{n} \nu_\ell = 1$. *Here, n represents a balancing coefficient.*

Note that BNEHA reduces to BNEWA if $w = \left( \dfrac{1}{n}, \dfrac{1}{n}, ....., \dfrac{1}{n} \right)^T$ or the BNEOWA operator if $v = \left( \dfrac{1}{n}, \dfrac{1}{n}, ..., \dfrac{1}{n} \right)$.

**Theorem 10.** *The (BNEHA) operator precedesa BN number with*

$$BNEHA_{w,v}\left( u_1, u_2, ....., u_n \right) =$$

$$\left( \frac{\dot{\mathcal{B}}_1 - \dot{\mathcal{B}}_2}{\dot{\mathcal{B}}_1 + \dot{\mathcal{B}}_2}, \frac{2\dot{\zeta}_1}{\dot{\zeta}_2 + \dot{\zeta}_1}, \frac{2\dot{f}_1}{\dot{f}_2 + \dot{f}_1}, \frac{-2\dot{\mathcal{B}}_3}{\dot{\mathcal{B}}_4 + \dot{\mathcal{B}}_3}, -\frac{\dot{\zeta}_3 - \dot{\zeta}_4}{\dot{\zeta}_3 + \dot{\zeta}_4}, -\frac{\dot{f}_3 - \dot{f}_4}{\dot{f}_3 + \dot{f}_4} \right) \qquad (7)$$

*where,*

$$\dot{\mathcal{B}}_1 = \prod_{\ell=1}^{n}\left(1 + \dot{\mathfrak{I}}^+_{\rho(\ell)}\right)^{v_\ell}, \quad \dot{\mathcal{B}}_2 = \prod_{\ell=1}^{n}\left(1 - \dot{\mathfrak{I}}^+_{\rho(\ell)}\right)^{v_\ell}, \quad \dot{\mathcal{B}}_3 = \prod_{\ell=1}^{n}\left|\dot{\mathfrak{I}}^-_{\rho(\ell)}\right|^{v_\ell}, \quad \dot{\mathcal{B}}_4 = \prod_{\ell=1}^{n}\left(2 + \dot{\mathfrak{I}}^-_{\rho(\ell)}\right)^{v_\ell},$$

$$\dot{\zeta}_1 = \prod_{\ell=1}^{n}\left(\dot{\mathfrak{i}}^+_{\rho(\ell)}\right)^{v_\ell}, \quad \dot{\zeta}_2 = \prod_{\ell=1}^{n}\left(2 - \dot{\mathfrak{i}}^+_{\rho(\ell)}\right)^{v_\ell}, \quad \dot{\zeta}_3 = \prod_{\ell=1}^{n}\left(1 + \left|\dot{\mathfrak{i}}^-_{\rho(\ell)}\right|\right)^{v_\ell}, \quad \dot{\zeta}_4 = \prod_{\ell=1}^{n}\left(1 + \dot{\mathfrak{i}}^-_{\rho(\ell)}\right)^{v_\ell},$$

$$\dot{f}_1 = \prod_{\ell=1}^{n}\left(\dot{f}^+_{\rho(\ell)}\right)^{v_\ell}, \quad \dot{f}_2 = \prod_{\ell=1}^{n}\left(2 - \dot{f}^+_{\rho(\ell)}\right)^{v_\ell}, \quad \dot{f}_3 = \prod_{\ell=1}^{n}\left(1 + \left|\dot{f}^-_{\rho(\ell)}\right|\right)^{v_\ell}, \quad \dot{f}_4 = \prod_{\ell=1}^{n}\left(1 + \dot{f}^-_{\rho(\ell)}\right)^{v_\ell}.$$

*where* $w = \left(w_1, w_2, w_3, ..., w_{n-1}, w_n\right)$ *is a weighting vector for* $u_\ell$, *by means of* $w_\ell \in \left[0, \ 1\right]$,

$\displaystyle\sum_{\ell=1}^{n} w_\ell = 1$ , *and* $\dot{u}_{\rho(\ell)}$ *is the* $\ell$ *-th biggest component of BN argument*

$\dot{u}_\ell \left( \dot{u}_\ell = (nv_\ell)u_\ell, \ell = 1, 2, ..., n \right)$. *Moreover,* $v = \left(v_1, v_2, ..., v_n\right)$ *is the weighting vector of*

*BN argument* $u_\ell$, *with* $v_\ell \in \left[0, \ 1\right], \displaystyle\sum_{\ell=1}^{n} v_\ell = 1$.

**Proof.** The theorem is straightforward.□

## 4. Bipolar Neutrosophic Einstein Geometric AOs

In the current section, we extend our study to bipolar neutrosophic Einstein geometric AOs, such as BNEWG, BNEOWG and BNHA, as well as their required properties.

*4.1. Bipolar Neutrosophic EinsteinWeighted Geometric AO*

Let $u_\ell = \left( \mathfrak{I}_\ell^+, \mathrm{I}_\ell^+, f_\ell^+, \mathfrak{I}_\ell^-, \mathrm{I}_\ell^-, f_\ell^- \right) (\ell = 1, 2, ..., n)$ represent a family of bipolar neutrosophic values.

**Definition 19**. *The (BNEWG) bipolar neutrosophic Einstein weighted geometric operator is defined as:*

$$BNEWG_v \left( u_1, u_2, u_3, ..., u_n \right) = \overset{n}{\underset{\ell=1}{\otimes}} \left( v_\ell u_\ell \right) = v_1 u_1 \otimes v_2 u_2 \otimes ....... \otimes v_n u_n \tag{8}$$

*where* $v = \left( v_1, v_2, v_3, ..., v_n \right)^T$ *is a weighting vector of* $u_\ell$, *with* $v_\ell > 0$ *and* $\sum_{\ell=1}^{n} v_\ell = 1$.

**Theorem 11.** *The (BNEWG) operators return a BN value with*

$$BNEWG_v \left( u_1, u_2, u_3, ..., u_n \right) =$$
$$\left( \frac{2\mathcal{B}_{\bar{1}}}{\mathcal{B}_{\bar{2}} + \mathcal{B}_{\bar{1}}}, \frac{\zeta_{\bar{1}} - \zeta_{\bar{2}}}{\zeta_{\bar{1}} + \zeta_{\bar{2}}}, \frac{f_{\bar{1}} - f_{\bar{2}}}{f_{\bar{1}} + f_{\bar{2}}}, -\frac{\mathcal{B}_{\bar{3}} - \mathcal{B}_{\bar{4}}}{\mathcal{B}_{\bar{3}} + \mathcal{B}_{\bar{4}}}, \frac{-2\zeta_{\bar{3}}}{\zeta_{\bar{4}} + \zeta_{\bar{3}}}, \frac{-2f_{\bar{3}}}{f_{\bar{4}} + f_{\bar{3}}} \right) \tag{9}$$

*where*

$$\mathcal{B}_{\bar{1}} = \prod_{\ell=1}^{n} \left( \mathfrak{I}_\ell^+ \right)^{v_\ell}, \quad \mathcal{B}_{\bar{2}} = \prod_{\ell=1}^{n} \left( 2 - \mathfrak{I}_\ell^+ \right)^{v_\ell}, \quad \mathcal{B}_{\bar{3}} = \prod_{\ell=1}^{n} \left( 1 + \left| \mathfrak{I}_\ell^- \right| \right)^{v_\ell}, \quad \mathcal{B}_{\bar{4}} = \prod_{\ell=1}^{n} \left( 1 + \mathfrak{I}_\ell^- \right)^{v_\ell},$$

$$\zeta_{\bar{1}} = \prod_{\ell=1}^{n} \left( 1 + \mathrm{I}_\ell^+ \right)^{v_\ell}, \quad \zeta_{\bar{2}} = \prod_{\ell=1}^{n} \left( 1 - \mathrm{I}_\ell^+ \right)^{v_\ell}, \quad \zeta_{\bar{3}} = \prod_{\ell=1}^{n} \left| \mathrm{I}_\ell^- \right|^{v_\ell}, \quad \zeta_{\bar{4}} = \prod_{\ell=1}^{n} \left( 2 + \mathrm{I}_\ell^- \right)^{v_\ell},$$

$$f_{\bar{1}} = \prod_{\ell=1}^{n} \left( 1 + f_\ell^+ \right)^{v_\ell}, \quad f_{\bar{2}} = \prod_{\ell=1}^{n} \left( 1 - f_\ell^+ \right)^{v_\ell}, \quad f_{\bar{3}} = \prod_{\ell=1}^{n} \left| f_\ell^- \right|^{v_\ell}, \quad f_{\bar{4}} = \prod_{\ell=1}^{n} \left( 2 + f_\ell^- \right)^{v_\ell}.$$

*where* $v = \left( v_1, v_2, v_3, ..., v_n \right)^T$ *is a weighting vector of* $u_\ell$, *with* $v_\ell > 0$ *as well as*

$$\sum_{\ell=1}^{n} v_\ell = 1.$$

**Proof.** The proof follows from Theorem 1.□

**Theorem 12.** *(Idempotency) Let* $u_\ell = \left( \mathfrak{I}_\ell^+, \mathrm{I}_\ell^+, f_\ell^+, \mathfrak{I}_\ell^-, \mathrm{I}_\ell^-, f_\ell^- \right) (\ell = 1, 2, 3, ..., n)$ *be a set of equal BNVs, that is,* $u_\ell = u$

$$BNEWG_v \left( u_1, u_2, u_3, ..., u_n \right) = u$$

**Proof.** Since

$$BNEWG_v \left( u_1, u_2, ....., u_n \right)$$
$$= \overset{n}{\underset{\ell=1}{\otimes}} \left( v_\ell u_\ell \right) = \overset{n}{\underset{\ell=1}{\otimes}} \left( v_\ell \right) u = \left( u \right) \sum_{\ell=1}^{n} v_\ell = u$$

The proof is complete.□

**Theorem 13.** *(Bounded) Suppose that* $u^- = \min_{\ell} u_\ell$, $u^+ = \max_{\ell} u_\ell$, *then*

$$u^{-} \leq BNEWG_{\nu}\left(u_{1}, u_{2}, \ldots, u_{n}\right) \leq u^{+}$$

**Proof.** Let

$$BNEWG_{\nu}\left(u_{1}, u_{2}, \ldots, u_{n}\right) = u\left(\Im^{+}, I^{+}, f^{+}, \Im^{-}, I^{-}, f^{-}\right)$$

Then

$$s\left(u^{-}\right) \leq s\left(BNEWG \text{ operator}\right)$$

$$s\left(u^{+}\right) \geq s\left(BNEWG \text{ operator}\right)$$

Combining both equations, we have

$$u^{-} \leq BNEWG_{\nu}\left(u_{1}, u_{2}, \ldots, u_{n}\right) \leq u^{+}$$

Thus, the proof is complete.□

**Theorem 14.** *(Monotonicity) Let* $u_{\ell} = \left(\Im_{\ell}^{+}, I_{\ell}^{+}, f_{\ell}^{+}, \Im_{\ell}^{-}, I_{\ell}^{-}, f_{\ell}^{-}\right)\left(\ell = 1, 2, \ldots, n\right)$ *and* $u_{\ell}' = \left(\Im_{\ell}'^{+}, I_{\ell}'^{+}, f_{\ell}'^{+}, \Im_{\ell}'^{-}, I_{\ell}'^{-}, f_{\ell}'^{-}\right)\left(\ell = 1, 2, \ldots, n\right)$ *be a collection of BNVs. If* $u_{\ell} \leq u_{\ell}'$*, then*

$$BNEWG_{\nu}\left(u_{1}, u_{2}, \ldots, u_{n}\right) \leq BNEWG_{\nu}\left(u_{1}', u_{2}', \ldots, u_{n}'\right)$$

**Proof.** Assume that

$$BNEWG_{\nu}\left(u_{1}, u_{2}, \ldots, u_{n}\right) = \overset{n}{\underset{\ell=1}{\otimes}}\left(\nu_{\ell} u_{\ell}\right)$$

and

$$BNEWG_{\nu}\left(u_{1}', u_{2}', \ldots, u_{n}'\right) = \overset{n}{\underset{\ell=1}{\otimes}}\left(\nu_{\ell} u_{\ell}'\right)$$

Since $u_{\ell} \leq u_{\ell}'$, then

$$\overset{n}{\underset{\ell=1}{\otimes}}\left(\nu_{\ell} u_{\ell}\right) \leq \overset{n}{\underset{\ell=1}{\otimes}}\left(\nu_{\ell} u_{\ell}'\right)$$

Thus, the proof is complete.□

*4.2. BN Einstein OrderedWeighted Geometric Aggregation Operators*

**Definition 20.** *The (BNEOWG) BN Einstein ordered weighted geometric AO is stated as:*

$$BNEOWG_v\left(u_1, u_2, ..., u_{n-1},\ u_n\right) = \overset{n}{\underset{\ell=1}{\otimes}}\left(v_\ell u_{\sigma(\ell)}\right) = v_1 u_{\sigma(1)} \otimes v_2 u_{\sigma(2)} \otimes v_3 u_{\sigma(3)} \otimes ...... \otimes v_\ell u_{\sigma(\ell)}, \quad (10)$$

*where* $\left(\sigma(1), ..., \sigma(n-1), \sigma(n)\right)$ *represents permutation* $(1, 2, 3, ..., n-1, n)$ *, that is,* $u_{\sigma(\ell-1)} \geq u_{\sigma(\ell)}$ *and* $(j = 2, ..., n-1, n)$, *and* $v = \left(v_1, v_2, v_3, ..., v_n\right)^T$ *is a weighting vector of* $u_\ell$*, with* $v_\ell > 0$ *and* $\sum_{\ell=1}^{n} v_\ell = 1.$

**Theorem 15.** *The (BNEOWG) operators return a BN value with*

$$BNEOWG_v\left(u_1, u_2, ....., u_n\right) =$$
$$\left(\frac{2\mathcal{B}_{\partial\tilde{1}}}{\mathcal{B}_{\partial\tilde{2}} + \mathcal{B}_{\partial\tilde{1}}}, \frac{\zeta_{\partial\tilde{1}} - \zeta_{\partial\tilde{2}}}{\zeta_{\partial\tilde{1}} + \zeta_{\partial\tilde{2}}}, \frac{f_{\partial\tilde{1}} - f_{\partial\tilde{2}}}{f_{\partial\tilde{1}} + f_{\partial\tilde{2}}}, -\frac{\mathcal{B}_{\partial\tilde{3}} - \mathcal{B}_{\partial\tilde{4}}}{\mathcal{B}_{\partial\tilde{3}} + \mathcal{B}_{\partial\tilde{4}}}, \frac{-2\zeta_{\partial\tilde{3}}}{\zeta_{\partial\tilde{4}} + \zeta_{\partial\tilde{3}}}, \frac{-2f_{\partial\tilde{3}}}{f_{\partial\tilde{4}} + f_{\partial\tilde{3}}}\right) \quad (11)$$

*where,*

$$\mathcal{B}_{\partial\tilde{1}} = \prod_{j=1}^{n}\left(\mathfrak{I}_{\sigma(j)}^{+}\right)^{\omega_j}, \quad \mathcal{B}_{\partial\tilde{2}} = \prod_{j=1}^{n}\left(2 - \mathfrak{I}_{\sigma(j)}^{+}\right)^{\omega_j}, \quad \mathcal{B}_{\partial\tilde{3}} = \prod_{j=1}^{n}\left(1 + \left|\mathfrak{I}_{\sigma(j)}^{-}\right|\right)^{\omega_j}, \quad \mathcal{B}_{\partial\tilde{4}} = \prod_{j=1}^{n}\left(1 + \mathfrak{I}_{\sigma(j)}^{-}\right)^{\omega_j},$$

$$\zeta_{\partial\tilde{1}} = \prod_{j=1}^{n}\left(1 + \mathrm{I}_{\sigma(j)}^{+}\right)^{\omega_j}, \quad \zeta_{\partial\tilde{2}} = \prod_{j=1}^{n}\left(1 - \mathrm{I}_{\sigma(j)}^{+}\right)^{\omega_j}, \quad \zeta_{\partial\tilde{3}} = \prod_{j=1}^{n}\left|\mathrm{I}_{\sigma(j)}^{-}\right|^{\omega_j}, \quad \zeta_{\partial\tilde{4}} = \prod_{j=1}^{n}\left(2 + \mathrm{I}_{\sigma(j)}^{-}\right)^{\omega_j},$$

$$f_{\partial\tilde{1}} = \prod_{j=1}^{n}\left(1 + f_{\sigma(j)}^{+}\right)^{\omega_j}, \quad f_{\partial\tilde{2}} = \prod_{j=1}^{n}\left(1 - f_{\sigma(j)}^{+}\right)^{\omega_j}, \quad f_{\partial\tilde{3}} = \prod_{j=1}^{n}\left|f_{\sigma(j)}^{-}\right|^{\omega_j}, \quad f_{\partial\tilde{4}} = \prod_{j=1}^{n}\left(2 + f_{\sigma(j)}^{-}\right)^{\omega_j}.$$

*where* $\left(\sigma(1), \sigma(2), \sigma(3), ..., \sigma(n)\right)$ *represents a permutation of* $(1, 2, ..., n)$ *with* $u_{\sigma(\ell-1)} \geq u_{\sigma(\ell)}$ *and* $(\ell = 2, 3, ..., n)$, *and* $v = \left(v_1, v_2, v_3, ..., v_n\right)^T$ *is a weighting vector of* $u_\ell$*, with* $v_\ell > 0$ *and* $\sum_{\ell=1}^{n} v_\ell = 1.$

**Proof.** The theorem is simple to understand.□

**Theorem 16.** *(Commutativity) Let* $u_\ell = \left(\mathfrak{I}_\ell^{+}, \mathrm{I}_\ell^{+}, f_\ell^{+}, \mathfrak{I}_\ell^{-}, \mathrm{I}_\ell^{-}, f_\ell^{-}\right)$, *as well as* $u_\ell' = \left(\mathfrak{I}_\ell'^{+}, \mathrm{I}_\ell'^{+}, f_\ell'^{+}, \mathfrak{I}_\ell'^{-}, \mathrm{I}_\ell'^{-}, f_\ell'^{-}\right)$, *where* $\ell \in Z$ *represents two BNNs.*

$$BNEOWG_\varpi\left(u_1, u_2, ..., u_n\right) = BNEOWG_\varpi\left(u_1', u_2', ..., u_n'\right)$$

*where* $\left(u_1, u_2, ..., u_n\right)$ *is any permutation of* $\left(u_1', u_2', ..., u_n'\right)$.

**Proof.** Assume that

$$BNEOWG_v\left(u_1, u_2, ..., u_n\right) = \overset{n}{\underset{\ell=1}{\otimes}}\left(v_\ell u_{\rho(\ell)}\right)$$

and

$$BNEOWG_v\left(u'_1, u'_2, ..., u'_n\right) = \overset{n}{\underset{\ell=1}{\otimes}}\left(v_\ell u'_{\rho(\ell)}\right)$$

Since $\left(u_1, u_2, ..., u_n\right)$ is any permutation of $\left(u'_1, u'_2, ..., u'_n\right)$, then

$$\overset{n}{\underset{\ell=1}{\otimes}}\left(v_\ell u_{\rho(\ell)}\right) = \overset{n}{\underset{\ell=1}{\otimes}}\left(v_\ell u'_{\rho(\ell)}\right)$$

The proof is complete.□

**Theorem 17.** *(Idempotency) Let* $u_\ell = \left(\mathfrak{I}^+_\ell, I^+_\ell, f^+_\ell, \mathfrak{I}^-_\ell, I^-_\ell, f^-_\ell\right)\left(\ell = 1, 2, ..., n\right)$ *represent a set of all equal BNVs, that is,* $u_\ell = u$

$$BNEOWG_v\left(u_1, u_2, u_3, ..., u_{n-1}, u_n\right) = u$$

**Theorem 18.** *(Bounded) Consider* $u^- = \min_\ell u_\ell$, $u^+ = \max_\ell u_\ell$ *; then,*

$$u^- \le BNEOWG_v\left(u_1, u_2, ...., u_n\right) \le u^+$$

**Theorem 19.** *(Monotonicity) Let* $u_\ell = \left(\mathfrak{I}^+_\ell, I^+_\ell, f^+_\ell, \mathfrak{I}^-_\ell, I^-_\ell, f^-_\ell\right)\left(\ell = 1, 2, ..., n\right)$ *and* $u'_\ell = \left(\mathfrak{I}'^+_\ell, I'^+_\ell, f'^+_\ell, \mathfrak{I}'^-_\ell, I'^-_\ell, f'^-_\ell\right)\left(\ell = 1, 2, ..., n\right)$ *be a collection of two BNVs. If* $u_\ell \le u'_\ell$,

$$BNEOWG_v\left(u_1, u_2, u_3, ..., u_n\right) \le BNEOWG_v\left(u'_1, u'_2, u'_3, ..., u'_n\right)$$

*4.3. BN Einstein HybridGeometric Aggregation Operators*

**Definition 21.***The BN Einstein hybrid geometric (BNEHG) aggregation operator is stated as follows:*

$$BNEHG_{w,v}\left(u_1, u_2, u_3, ..., u_{n-1}, u_n\right) = \overset{n}{\underset{\ell=1}{\otimes}}\left(v_\ell \dot{u}_{\sigma(\ell)}\right) = v_1\dot{u}_{\sigma(1)} \otimes v_2\dot{u}_{\sigma(2)} \otimes v_3\dot{u}_{\sigma(3)} \otimes ...... \otimes v_\ell\dot{u}_{\sigma(\ell)}, \tag{12}$$

*where* $w = \left(w_1, w_2, w_3, ..., w_{n-1}, w_n\right)$ *is a weighting vector of* $u_\ell$, *such that* $w_\ell \in \left[0, 1\right]$, $\sum_{\ell=1}^{n} w_\ell = 1$ *and* $\dot{u}_{\rho(\ell)}$ *represent the* $\ell$ *-th biggest component of the BN argument* $\dot{u}_\ell\left(\dot{u}_\ell = \left(nv_\ell\right)u_\ell, \ell = 1, 2, ..., n\right)$, *and* $v = \left(v_1, v_2, ..., v_n\right)$ *is the weighting vector of BN arguments* $u_\ell$,*along with* $v_\ell \in \left[0, 1\right]$, $\sum_{\ell=1}^{n} v_\ell = 1$.

Note that BNEHG reduces to BNEWG if $w = \left(\dfrac{1}{n}, \dfrac{1}{n}, \ldots, \dfrac{1}{n}\right)^T$ or the BNEOWG operator if $v = \left(\dfrac{1}{n}, \dfrac{1}{n}, \ldots, \dfrac{1}{n}\right)$.

**Theorem 20.** *The (BNEHG) operator gives in return a BNV with*

$$BNEHG_{w,v}\left(u_1, u_2, u_3, \ldots, u_n\right) =$$

$$\left(\frac{2\dot{\mathcal{B}}_{\dot{1}}}{\dot{\mathcal{B}}_{\dot{2}} + \dot{\mathcal{B}}_{\dot{1}}}, \frac{\dot{\zeta}_{\dot{1}} - \dot{\zeta}_{\dot{2}}}{\dot{\zeta}_{\dot{1}} + \dot{\zeta}_{\dot{2}}}, \frac{\dot{f}_{\dot{1}} - \dot{f}_{\dot{2}}}{\dot{f}_{\dot{1}} + \dot{f}_{\dot{2}}}, -\frac{\dot{\mathcal{B}}_{\dot{3}} - \dot{\mathcal{B}}_{\dot{4}}}{\dot{\mathcal{B}}_{\dot{3}} + \dot{\mathcal{B}}_{\dot{4}}}, \frac{-2\dot{\zeta}_{\dot{3}}}{\dot{\zeta}_{\dot{4}} + \dot{\zeta}_{\dot{3}}}, \frac{-2\dot{f}_{\dot{3}}}{\dot{f}_{\dot{4}} + \dot{f}_{\dot{3}}}\right) \qquad (13)$$

*where*

$$\dot{\mathcal{B}}_{\dot{1}} = \prod_{j=1}^{n}\left(\dot{\mathfrak{I}}_{\sigma(j)}^{+}\right)^{\omega_j}, \quad \dot{\mathcal{B}}_{\dot{2}} = \prod_{j=1}^{n}\left(2 - \dot{\mathfrak{I}}_{\sigma(j)}^{+}\right)^{\omega_j}, \quad \dot{\mathcal{B}}_{\dot{3}} = \prod_{j=1}^{n}\left(1 + \left|\dot{\mathfrak{I}}_{\sigma(j)}^{-}\right|\right)^{\omega_j}, \quad \dot{\mathcal{B}}_{\dot{4}} = \prod_{j=1}^{n}\left(1 + \dot{\mathfrak{I}}_{\sigma(j)}^{-}\right)^{\omega_j},$$

$$\dot{\zeta}_{\dot{1}} = \prod_{j=1}^{n}\left(1 + \dot{\mathbf{I}}_{\sigma(j)}^{+}\right)^{\omega_j}, \quad \dot{\zeta}_{\dot{2}} = \prod_{j=1}^{n}\left(1 - \dot{\mathbf{I}}_{\sigma(j)}^{+}\right)^{\omega_j}, \quad \dot{\zeta}_{\dot{3}} = \prod_{j=1}^{n}\left|\dot{\mathbf{I}}_{\sigma(j)}^{-}\right|^{\omega_j}, \quad \dot{\zeta}_{\dot{4}} = \prod_{j=1}^{n}\left(2 + \dot{\mathbf{I}}_{\sigma(j)}^{-}\right)^{\omega_j},$$

$$\dot{f}_{\dot{1}} = \prod_{j=1}^{n}\left(1 + \dot{f}_{\sigma(j)}^{+}\right)^{\omega_j}, \quad \dot{f}_{\dot{2}} = \prod_{j=1}^{n}\left(1 - \dot{f}_{\sigma(j)}^{+}\right)^{\omega_j}, \quad \dot{f}_{\dot{3}} = \prod_{j=1}^{n}\left|\dot{f}_{\sigma(j)}^{-}\right|^{\omega_j}, \quad \dot{f}_{\dot{4}} = \prod_{j=1}^{n}\left(2 + \dot{f}_{\sigma(j)}^{-}\right)^{\omega_j}.$$

*where $w = \left(w_1, w_2, w_3, \ldots, w_{n-1}, w_n\right)$ is a weighting vector of $u_{\ell}$, such that $w_{\ell} \in \left[0,\ 1\right]$,*

$\sum_{\ell=1}^{n} w_{\ell} = 1$ *and $\dot{u}_{\rho(\ell)}$ represent the $\ell$-th biggest component of the BN argument*

$\dot{u}_{\ell}\left(\dot{u}_{\ell} = \left(n v_{\ell}\right) u_{\ell}, \ell = 1, 2, \ldots, n\right)$, *and $v = \left(v_1, v_2, \ldots, v_n\right)$ is the weighting vector of BN ar-*

*guments $u_{\ell}$, along with $v_{\ell} \in [0,1]$, $\sum_{\ell=1}^{n} v_{\ell} = 1$. Here, n represents a balancing coefficient.*

**Proof.** The theorem is straightforward.□

## 5. Multi-Criteria Group DM Problem Based on BN Einstein Aggregation Operators

This section includes a multi-criteria application based on BN Einstein aggregation operators (AOs), along with crisp numbers serving as attributes' weights and BN numbers serving as attributes' values.

*5.1. Algorithm*

Consider $G = \left\{G_1, G_2, \ldots, G_m\right\}$ to represent a collection of finite $m$ alternatives, $L = \left\{L_1, L_2, L_3, \ldots, L_n\right\}$ to represent a collection of finite $n$ attributes and $D = \left\{D_1, D_2, D_3, \ldots, D_k\right\}$ to be a finite number of $k$ decision-makers. Suppose $v = \left(v_1, v_2, \ldots, v_n\right)^T$ is the weighted vector for decision-makers $\overline{D^s}\left(s = 1, 2, 3, \ldots, k-1, k\right)$, with $v_{\ell} \in \left[0,\ 1\right]$ and $\sum_{\ell=1}^{n} v_{\ell} = 1$. Suppose $w = \left(w_1, w_2, w_3, \ldots, w_n\right)^T$ is the weighted vector representing the set of attributes $L = \left\{L_1, L_2, L_3, \ldots, L_n\right\}$, with $w_{\ell} \in [0,1]$ and

$\sum_{\ell=1}^{n} w_{\ell} = 1.$ The decision-maker assesses an option based on a set of criteria, the values of which are defined by BNVs. Let $u_{\hat{i}\hat{j}}^{(s)} = \left[ \left( \Im_{\hat{i}\hat{j}}^{+}, I_{\hat{i}\hat{j}}^{+}, f_{\hat{i}\hat{j}}^{+}, \Im_{\hat{i}\hat{j}}^{-}, I_{\hat{i}\hat{j}}^{-}, f_{\hat{i}\hat{j}}^{-} \right) \right]_{m \times n}$ be the decision matrices provided by an expert decision-maker, and $u_{\hat{i}\hat{j}}^{(s)}$ be a BNN for $L_{\hat{i}}$ attribute associated with alternatives. We have $\Im_{\hat{i}\hat{j}}^{+}, I_{\hat{i}\hat{j}}^{+}, f_{\hat{i}\hat{j}}^{+}, \Im_{\hat{i}\hat{j}}^{-}, I_{\hat{i}\hat{j}}^{-}$ and $f_{\hat{i}\hat{j}}^{-} \in \left[ 0, 1 \right]$ with condition $0 \leq \Im_{\hat{i}\hat{j}}^{+} + I_{\hat{i}\hat{j}}^{+} + f_{\hat{i}\hat{j}}^{+} + \Im_{\hat{i}\hat{j}}^{-} + I_{\hat{i}\hat{j}}^{-} + f_{\hat{i}\hat{j}}^{-} \leq 6$, where $\hat{i} = 1, 2, 3, ..., m-1, m$ and $\hat{j} = 1, 2, 3, ..., n$. (Figure 1)

**Step 1:** Matrix construction (Table:1,2,3) for decision-making

$$\overline{D^s} = \left[ u_{\hat{i}\hat{j}}^{(s)} \right]_{m \times n} \left( s = 1, 2, 3, ..., k-1, k \right).$$

**Step 2:** Computing $BNEWA_v \left( r_{\hat{i}1}, r_{\hat{i}2}, ....., r_{\hat{i}n} \right)$ (Table:4) for every $\hat{i} = 1, 2, 3, ..., m$.

$r_{\hat{i}} = \left( \Im_{\hat{i}}^{+}, I_{\hat{i}}^{+}, f_{\hat{i}}^{+}, \Im_{\hat{i}}^{-}, I_{\hat{i}}^{-}, f_{\hat{i}}^{-} \right) =$

$BNEWA_v \left( r_{\hat{i}1}, r_{\hat{i}2}, ....., r_{\hat{i}n} \right) = \overset{n}{\underset{\ell=1}{\oplus}} \left( v_{\ell} r_{\hat{i}\ell} \right) =$

$\left( \dfrac{\mathcal{B}_1 - \mathcal{B}_2}{\mathcal{B}_1 + \mathcal{B}_2}, \dfrac{2\zeta_1}{\zeta_2 + \zeta_1}, \dfrac{2f_1}{f_2 + f_1}, \dfrac{-2\mathcal{B}_3}{\mathcal{B}_4 + \mathcal{B}_3}, -\dfrac{\zeta_3 - \zeta_4}{\zeta_3 + \zeta_4}, -\dfrac{f_3 - f_4}{f_3 + f_4} \right)$

**Step 3:** Computing the scores values $S(r_{\hat{i}})$.

**Step 4:** Various software systems are ranked for $BNEWA_v \left( u_{\hat{i}1}, u_{\hat{i}2}, ....., u_{\hat{i}n} \right)$ in accordance with their scores.

**Step 5:** Picking the best option.

*5.2. Illustrative Example*

We believe that a medicine business company should hire a professional manager. For this reason, the organization formsa working group with three decision-makers in weighting vectors $v = (0.1, 0.5, 0.4)^T$. When choosing the most knowledgeable management approach, there are various factors to evaluate, but in this situation, the committee just analyses the four criteriamentioned below, along with associated weighted vector $v = (0.1, 0.4, 0.2, 0.3)^T$. Four managers $G_j (j = 1, 2, 3, 4)$ will advance to the next round of the procedure after passing the first screening exam.

L₁: Salary;
L₂: Experience;
L₃: Working skill;
L₄: Dealing with public.

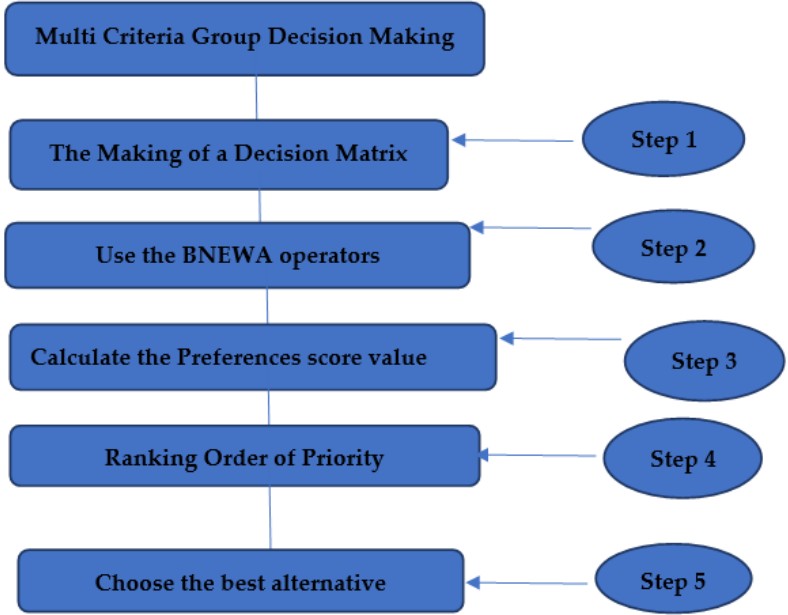

**Figure 1.** Flow Chart for MCGDM.

**Step 1.** Matrices construction for decisions.

**Table 1.** BN decision matrix by decision-maker $D_1$.

| | $L_1$ | $L_2$ | $L_3$ | $L_4$ |
|---|---|---|---|---|
| $G_1$ | (0.2,0.6,0.5,−0.3,−0.9,−0.6) | (0.4,0.6,0.7,−0.5,−0.4,−0.3) | (0.8,0.3,0.5,−0.6,−0.1,−0.9) | (0.5,0.4,0.6,−0.8,−0.5,−0.4) |
| $G_2$ | (0.5,0.7,0.3,−0.8,−0.5,−0.7) | (0.3,0.4,0.6,−0.8,−0.7,−0.6) | (0.4,0.6,0.9,−0.5,−0.4,−0.5) | (0.3,0.8,0.9,−0.1,−0.4−0.3) |
| $G_3$ | (0.5,0.7,0.8,−0.4,−0.7,−0.4) | (0.3,0.7,0.7,−0.5,−0.3,−0.2) | (0.5,0.3,0.4,−0.6,−0.7,−0.9) | (0.4,0.6,0.5,−0.3,−0.5,−0.8) |
| $G_4$ | (0.8,0.5,0.4,−0.7,−0.6,−0.5) | (0.2,0.4,0.5,−0.8,−0.6,−0.3) | (0.3,0.7,0.4,−0.5,−0.7,−0.5) | (0.9,0.4,0.6,−0.5,−0.4,−0.7) |

**Table 2.** BN decision matrix by decision-maker $D_2$.

| | $L_1$ | $L_2$ | $L_3$ | $L_4$ |
|---|---|---|---|---|
| $G_1$ | (0.4,0.6,0.5,−0.7,−0.4,−0.8) | (0.5,0.4,0.6,−0.8,−0.5,−0.7) | (0.4,0.6,0.5,−0.4,−0.8,−0.5) | (0.5,0.6,0.3,−0.4,−0.6,−0.8) |
| $G_2$ | (0.4,0.7,0.5,−0.6,−0.3,−0.9) | (0.4,0.7,0.8,−0.3,−0.5,−0.4) | (0.2,0.5,0.7,−0.6,−0.5,−0.4) | (0.8,0.4,0.2,−0.8,−0.1,−0.4) |
| $G_3$ | (0.6,0.3,0.6,−0.3,−0.7,−0.8) | (0.6,0.4,0.6,−0.7,−0.5,−0.8) | (0.6,0.3,0.2,−0.1,−0.4,−0.7) | (0.5,0.6,0.7,−0.3,−0.5,−0.6) |
| $G_4$ | (0.2,0.3,0.4,−0.7,−0.6,−0.8) | (0.8,0.5,0.4,−0.7,−0.4,−0.6) | (0.8,0.4,0.5,−0.7,−0.5,−0.1) | (0.6,0.5,0.8,−0.7,−0.6,−0.4) |

**Table 3.** BN decision matrix by decision-maker $D_3$.

| | $L_1$ | $L_2$ | $L_3$ | $L_4$ |
|---|---|---|---|---|
| $G_1$ | (0.5,0.6,0.4,−0.7,−0.4,−0.3) | (0.2,0.5,0.6,−0.4,−0.7,−05) | (0.5,0.7,0.2,−0.9,−0.5,−0.3) | (0.5,0.6,0.3,−0.7,−0.5,−0.2) |
| $G_2$ | (0.9,0.2,0.4,−0.5,−0.4,−0.8) | (0.5,0.1,0.2,−0.9,−0.6,−0.4) | (0.1,0.4,0.8,−0.6,−0.5,−0.3) | (0.6,0.5,0.3,−0.9,−0.3,−0.5) |
| $G_3$ | (0.4,0.5,0.6,−0.1,−0.6,−0.5) | (0.3,0.4,0.8,−0.5,−0.4,−0.3) | (0.4,0.6,0.3,−0.4,−0.5,−0.3) | (0.5,0.4,0.9,−0.5,−0.4,−0.7) |
| $G_4$ | (0.1,0.4,0.5,−0.4,−0.8,−0.7) | (0.4,0.3,0.6,−0.2,−0.7,−0.5) | (0.7,0.5,0.6,−0.4,−0.3−0.9) | (0.1,0.5,0.7,−0.5,−0.8,−0.3) |

**Step 2:** Compute $BNEWA_v\left(r_{\tilde{i}1},r_{\tilde{i}2},\dots,r_{\tilde{i}n}\right)$

$$r_{\tilde{i}}=\left(\mathfrak{I}_{\tilde{i}}^{+},I_{\tilde{i}}^{+},f_{\tilde{i}}^{+},\mathfrak{I}_{\tilde{i}}^{-},I_{\tilde{i}}^{-},f_{\tilde{i}}^{-}\right)=$$

$$BNEWA_v\left(r_{\tilde{i}1},r_{\tilde{i}2},\dots,r_{\tilde{i}n}\right)=\overset{n}{\underset{\ell=1}{\oplus}}\left(v_\ell r_{i\ell}\right)=$$

$$\left(\frac{\mathcal{B}_1-\mathcal{B}_2}{\mathcal{B}_1+\mathcal{B}_2},\frac{2\zeta_1}{\zeta_2+\zeta_1},\frac{2f_1}{f_2+f_1},\frac{-2\mathcal{B}_3}{\mathcal{B}_4+\mathcal{B}_3},-\frac{\zeta_3-\zeta_4}{\zeta_3+\zeta_4},-\frac{f_3-f_4}{f_3+f_4}\right)$$

**Table 4.** A collective BN decision matrix R.

|  | $L_1$ | $L_2$ |
|---|---|---|
| $G_1$ | (0.4234,0.6000,0.4581,−0.6501,−0.4842,−0.6306) | (0.3783,0.4568,0.6096,−0.5917,−0.5810,−0.5943) |
| $G_2$ | (0.6940,0.4453,0.4360,−0.5766,−0.3620,−0.8517) | (0.4219,0.3288,0.4750,−0.5426,−0.5640,−0.4224) |
| $G_3$ | (0.5161,0.4057,0.6187,−0.2024,−0.6627,−0.6704) | (0.4632,0.4251,0.6870,−0.5952,−0.4423,−0.6001) |
| $G_4$ | (0.2462,0.3555,0.4381,−0.5675,−0.6938,−0.7403) | (0.6286,0.4014,0.4839,−0.4527,−0.5567,−0.5351) |
|  | $L_3$ | $L_4$ |
| $G_1$ | (0.4940,0.6011,0.3536,−0.5954,−0.6522,−0.4973) | (0.5000,0.5776,0.3233,−0.5453,−0.5520,−0.5868) |
| $G_2$ | (.1818,0.4670,0.7589,−0.5895,−0.4905,−0.3718) | (0.6950,0.4726,0.2807,−0.7190,−0.2130,−0.4321) |
| $G_3$ | (.5161,0.4021,0.2533,−0.2160,−0.4764,−0.6073) | (0.4905,0.5135,0.7552,−0.3708,−0.4614,−0.6659) |
| $G_4$ | (.7293,0.4649,0.5274,−0.5482,−0.4504,−0.6005) | (0.4884,0.4893,0.7387,−0.5952,−0.6796,−0.3989) |

**Step 3:** Now, compute the scoring function of

$$r_1 = \left(0.3029, 0.5336, 0.4445, -0.5839, -0.4416, -0.4277\right),$$

$$r_2 = \left(0.2934, 0.4068, 0.4501, -0.6059, -0.3806, -0.3667\right),$$

$$r_3 = \left(0.3556, 0.4435, 0.5882, -0.3867, -0.3570, -0.4616\right),$$

$$r_4 = \left(0.4670, 0.4341, 0.5578, -0.5236, -0.4085, -0.4405\right),$$

$$S\left(r_i\right) = \frac{1}{6}\left(\Im^+ + 1 - I^+ + 1 - f^+ + 1 + \Im^- - I^- - f^-\right),$$

$$S(r_1) = .4350, S(r_2) = 0.4296, S(r_3) = .4593, S(r_4) = 0.4668$$

**Step 4:** We have arrived at a conclusion by calculating the scores:

$$G_4 \succ G_3 \succ G_1 \succ G_2$$

**Step 5:** The best option is $G_4$.

## 6. Comparison

Different researchers have used a wide range of DM approaches so far. Chen et al. [28] used FSs, Atanassov [2] used IFSs, Zavadskas et al. [21] used NSs, Dubois et al. [34] used BFSsand Irfan et al. [25] used BNSs, among other study approaches.We employed Einstein operators to apply bipolarity to neutrosophic sets in this paper.

The aggregation operators in this study werebroader and more versatile, representing an advantage of our proposed method, which led us to determine that G4wasthe best manager for the job.

## 7. Conclusions

The goal of this research was to look at different BN aggregation operatorsdeveloped with the help of Einstein t-norms/t-conorms for multi-criteria community DM using BNVs as the criteria. The Einstein t-norm typically gives the same smooth approximations as the product and sumof an algebraic t-norm.Motivated by Einstein operations, we suggested bipolar neutrosophic Einstein aggregation operators for decision-making problems.To begin with, we looked at BN Einstein aggregation operators and the properties that they must have.Along with their attributes, these AOs were (BNEWA),(BNEOWA), (BNEHWA), (BNEWG), (BNEOWG)and (BNEHWG).Finally, we showed how to make multi-criteria decisions using such a framework. A descriptive case of manager selectionwas studied.The outcomes of the presentpaper show that the proposed approaches are accurate and practical when put into practice.We plan to extend the proposed approach to other domains and employ it in future research projects, such as pattern identification and risk analysis.

**Author Contributions:** Conceptualization, S.A.; Formal analysis, A.M.; Funding acquisition, A.R. and J.A.; Investigation, A.A. and J.A.; Methodology, M.J.; Project administration, F.A.; Validation, M.B.R.; Writing – review & editing, M.J.All authors have read and agreed to the published version of the manuscript.

**Funding:** This research received no external funding**.**

**Institutional Review Board Statement:** Not applicable

**Informed Consent Statement:** Not applicable

**Data Availability Statement:** Not applicable

**Conflicts of Interest:** The authors declare no conflict of interest.

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
