# Peer review of "Einstein Aggregation Operators under Bipolar Neutrosophic Environment with Applications in Multi-Criteria Decision-Making"

_applsci, doi:10.3390/app121910045_

Round 1
Reviewer 1 Report
The work is about the aggreation methods for MCDM in bipolar neutrosophic environment. 1.The English writing must be extensively improved.
2.The introduction should be reorganised. The highlights? The motivations? Some recent contributions should be reviewed.
3. The size of the equations should be readjusted.
4.What about the popular properties(symmetry, distributivity...) of the operations presented in Def. 15
5.You should present some introductions on Einstein operator. Why do you use the Einstein operator?
Author Response
ApplSci-1925966 (Reply to Review Report-1)
English language and style:
English is revised properly and changes are highlighted in the text.
Does the introduction provide sufficient background and include all relevant references?
There is an improvement of introduction shown in paragraph 4 and 5.
Are all the cited references relevant to the research?
Yes and some more references are added from 30 to 34.
Is the research design appropriate?
It is improved added few proofs, see theorem # 12,13,14.
Are the methods adequately described?
The methods are improved and highlighted in text.
Are the results clearly presented?
The results are improved shown in section 4.
Are the conclusions supported by the results?
The conclusion is improved also.
Comments and Suggestions for Authors:
- The English writing must be extensively improved:
There is improvement of English throughout the paper.
- The introduction should be reorganised. The highlights? The motivations? Some recent contributions should be reviewed.
There is an improvement of introduction shown in paragraph 4 and recent contribution added in paragraph 5.
- The size of the equations should be readjusted.
Size of equations re-adjusted.
- What about the popular properties (symmetry, distributivity....) of the operations presented in Def. 15
Theorem # 6 and 16 are added.
- You should present some introductions on Einstein operator. Why do you use the Einstein operator?
The material related to Einstein operations added in introduction paragraph 4.
Reviewer 2 Report
All results are new and well written.
I recommend for publication after some minor changes. Given below are some my suggestions to improve it.
1. In abstract paragraph is very long. It should have two parts one for motivation and other regarding work did in the paper.
2. In introduction, the text formatting issues in first paragraph. There are also space issues in first line of theorem 4 , 13 ,17 correct them.
3. Authors must remove all punctuation mistakes throughout the paper.
4. The references should be according to journal format. The authors must see all references carefully and make them uniform.
5. To improve literature review and historical background, in the introduction section these topics (but not limited to) should be discussed: soft sets, bipolar soft sets, linear Diophantine fuzzy sets, complex fuzzy sets, complex neutrosophic sets and bipolar complex fuzzy sets.
Author Response
ApplSci-1925966 (Reply to Review Report-2)
English language and style:
English is revised.
Does the introduction provide sufficient background and include all relevant references?
There is an improvement of introduction shown in paragraph 4 and paragraph 5.
Are all the cited references relevant to the research?
The references are improved
Are the conclusions supported by the results?
The conclusion is improved.
Comments and Suggestions for Authors:
- In abstract paragraph is very long. It should have two parts one for motivation and other regarding work did in the paper.
The abstract divided into two parts.
- In introduction, the text formatting issues in first paragraph. There are also space issues in first line of theorem 4 , 13 ,17 correct them.
These are corrected as desired.
- Authors must remove all punctuation mistakes throughout the paper.
Punctuation mistakes are removed throughout the paper.
- The references should be according to journal format. The authors must see all references carefully and make them uniform.
The references are converted according to journal format.
- To improve literature review and historical background, in the introduction section these topics (but not limited to) should be discussed: soft sets, bipolar soft sets, linear Diophantine fuzzy sets, complex fuzzy sets, complex neutrosophic sets and bipolar complex fuzzy sets.
The relevant literature is added in paragraph 5.
Reviewer 3 Report
The article is very good and interesting. I recommend it for publication.
Author Response
Thanks for the acceptance.
Round 2
Reviewer 1 Report
The authors cover the review comments well in the updated manuscript and revise the manuscript accordingly.